# Cellular 3D-reconstruction and analysis in the human cerebral cortex using automatic serial sections

Nick Y. Larsen [1,2,3,4✉], Xixia Li[5,6], Xueke Tan[5,6], Gang Ji[5,6], Jing Lin[7], Grazyna Rajkowska [8], Jesper Møller[2,9], Ninna Vihrs [9], Jon Sporring [2,7], Fei Sun [3,4,5,6,11] & Jens R. Nyengaard[1,2,3,10,11]

Techniques involving three-dimensional (3D) tissue structure reconstruction and analysis provide a better understanding of changes in molecules and function. We have developed AutoCUTS-LM, an automated system that allows the latest advances in 3D tissue reconstruction and cellular analysis developments using light microscopy on various tissues, including archived tissue. The workflow in this paper involved advanced tissue sampling methods of the human cerebral cortex, an automated serial section collection system, digital tissue library, cell detection using convolution neural network, 3D cell reconstruction, and advanced analysis. Our results demonstrated the detailed structure of pyramidal cells (number, volume, diameter, sphericity and orientation) and their 3D spatial organization are arranged in a columnar structure. The pipeline of these combined techniques provides a detailed analysis of tissues and cells in biology and pathology.

[1] Core Centre for Molecular Morphology, Section for Stereology and Microscopy, Department of Clinical Medicine, Aarhus University, Aarhus, Denmark. [2] Centre for Stochastic Geometry and Advanced Bioimaging, Aalborg University, Aarhus University and University of Copenhagen, Aalborg, Aarhus and Copenhagen, Denmark. [3] Sino-Danish Center for Education and Research, Aarhus, Denmark. [4] University of the Chinese Academy of Sciences, Beijing, China. [5] National Key Laboratory of Biomacromolecules, CAS Center for Excellence in Biomacromolecules, Institute of Biophysics, Chinese Academy of Sciences, Beijing, China. [6] Center for Biological Imaging, Institute of Biophysics, Chinese Academy of Sciences, Beijing, China. [7] Department of Computer Science, University of Copenhagen, Copenhagen, Denmark. [8] Department of Psychiatry and Human Behavior, University of Mississippi Medical Center, Jackson, MS, USA. [9] Department of Mathematical Sciences, Aalborg University, Aalborg, Denmark. [10] Department of Pathology, Aarhus University, Aarhus, Denmark. [11] These authors contributed equally: Fei Sun, Jens R. Nyengaard. ✉email: nylarsen@clin.au.dk

Life science aims at a better understanding of multiple biological functions, such as healthy organ development with cellular proliferation, migration and organization, tumor formation, and general pathology. Several techniques have been developed to study biological structure in 3D like serial block-face scanning electron microscopy (EM), focused ion beam scanning EM, serial-section transmission EM, automatic tape-collecting ultramicrotome-scanning EM, and many types of light microscopy with or without tissue clearing[1–4]. In life science, EM is widely used to explore the subcellular components, which are several orders of magnitude smaller than the spatial neuron networks. Clearing techniques can be very helpful in attempting to explain neuron networks in greater brain volumes, such as the adult mouse brain, in an entire state without disassembly[5,6]. However, some of the disadvantages for tissue clearance are that immunostaining of archival tissues is usually complicated as a result of the antigen masking due to formaldehyde protein cross-linking[7]. Furthermore, practicing immunostaining with tissue clearing remains difficult in human tissues due to factors such as inadequate antibody penetration depth, physicochemical properties, and tissue composition[7,8].

A typical human neuron has thousands of complex connections with neighboring neurons, which is essential for normal function, yet the organization of these neurons is still under debate[9]. The cellular organization in the human neocortex has been described as a local network of vertical columns containing neurons. Neurons with similar functions are grouped together and according to different theories, these cortical columns may contain smaller columns known as minicolumns, which are the smallest unit capable of processing information in the cerebral cortex[10,11]. Cortical columns are radially oriented cell bodies that span through the laminar pattern perpendicular to the pial surface and can be seen using regular Nissl preparations or another cell body-revealing histological procedures. The introduction of minicolumns was in response to studies of the patterning of apical dendrites of pyramidal cells with somata situated in layers II, III, and V[12,13]. Studies have attempted to characterize and analyze the morphology of minicolumns with a 2D computerized method designed to detect subtle differences among patient groups such as schizophrenia, autism, and Alzheimer[14–17]. As a result, much of our understanding of cellular organization is focused on 2D histological images, which could potentially misrepresent biological structures and malpositioning of cells in 3D-space.

This paper aims to create a method that is accessible to the broader science community and analyze 3D tissue organization through the use of archival tissue to make detailed inferences about pathologies. In the present study, we developed Automatic Collector of Ultrathin Sections for Light microscopy (AutoCUTS-LM) to measure the neuronal cell morphology and their spatial organization in 3D-space of archived tissue. This is accomplished by modifying and adjusting the original AutoCUTS, which was designed for scanning EM array tomography[18–20], to image archival human brain tissues (~20 years) in layer III of Brodmann Area 46 (BA46). BA46 was chosen since it involves working memory, attention and has been the subject of studies related to mental disorders like schizophrenia and depression[21–28]. Myelinated axon bundles are potentially cortical efferents that originated in layer II/III pyramidal cells as these bundles descend to the white matter[29]. Pyramidal neurons in layer III project to other cortical areas and play a key role in cortical and thalamic cortical circuits, and have been found to be the most affected layer by these disorders in BA46[30–32].

Here, we report the applied technical workflow that is able to uncover morphological properties of pyramidal neurons in human brain autopsy tissue: First, we identified BA46 and applied advanced sampling procedures of biopsies. After embedding biopsies in resin, the AutoCUTS-LM cut them automatically into semi-thin sections (300 and 400 $nm$), and collected them on tape. Sections were stained and mounted on glass slides, and a library for each sample was organized. The digitalization of sections was stored, aligned, and stacked as a volumetric structure. Then, we detected neurons using the UNetDense architecture. Finally, the 3D spatial arrangement and structural parameters of pyramidal neurons in layer III of BA46 were analyzed in three human brains applying recently developed methods.

Our findings present valuable insight into neuronal morphology and architecture by characterizing pyramidal neurons in 3D from old archived human brain tissue. We discovered that pyramidal cells are not randomly distributed but are clustered in small columnar structures, which may be relevant for understanding the formation and function of the cortical network.

## Results

**Sampling strategy and preparation of tissue.** A block of tissue from the dorsolateral prefrontal cortex that contained all of BA46 was removed from each brain[33], see Fig. 1. We used the MATLAB script we developed to delineate BA46 and sample two biopsies on a highly complex surface like the brain, with the second biopsy kept as a reserve. The cortical columns of neurons in the cortex could be successively extracted by a biopsy perpendicularly to the cortical surface; therefore, only neurons from gyri and not sulci were analyzed, illustrated with the yellow color in Fig. 1D. The sampling area was divided into four quarters to avoid any overlap or adjacent biopsies, see Fig. 1E, F.

The brain tissue samples were obtained by using a biopsy punch positioned on the two sampling points with a diameter of 1.5 $mm$ and a depth around 3–5 $mm$. This meant that a sample included all six layers of the cerebral cortex. The tissue samples were fixed in resin and not stained with osmium since osmium fixation reduces the contrast of sections during light microscopy, see Fig. S1.

Our strategy to successfully find neurons in the supragranular layers (I–III) and layer IV, region of interest (ROI), requires that the sample be positioned in such a manner that all six layers in the neocortex appear in every section during the AutoCUTS-LM procedure. Consequently, the biopsy was placed at the bottom of the embedding form such that the pial surface was perpendicular to the cutting direction, see Fig. S2. This orientation decreased the number of sections approximately two to three times relative to alternate orientations, thus shortened the time spent collecting and capturing images with the microscope. We used a light microscope to locate and measure the ROI by first examining and staining the outermost section of the block to delimit the area for collecting semi-thin sections (100–500 $nm$). We precisely trimmed the excess embedding resin around the sample to avoid wrinkles while cutting and removed layers V–VI, which resulted in an ROI of around 1.5 $mm^2$.

**Collection and preparation of serial brain sections on tape.** We have modified and adapted the automatic serial section technique, which was originally developed for scanning EM[18–20], so it operates for light microscopy. The main changes were the replacement of the double-sided conductive tape with a plasma cleaned transparent tape and mounting tape strips containing the sections on glass slides instead of on a silicon wafer. The transparent polyester tape was 7-$mm$ wide and was put through plasma treatment, which influenced the hydrophilicity of the tape. The hydrophilicity was found to not only reduce wrinkles of the brain sections produced during the collection process[20], but also

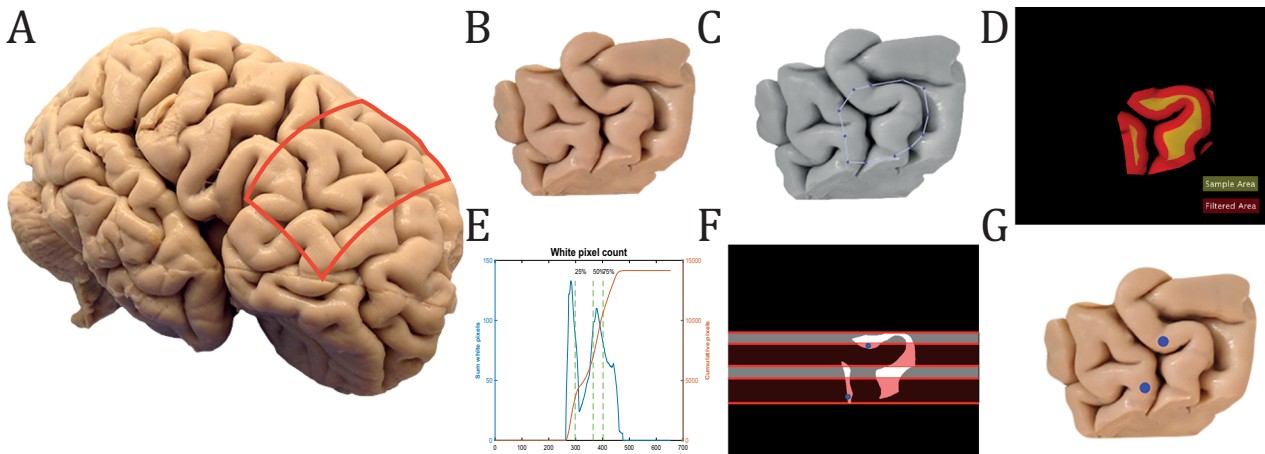

**Fig. 1 Sampling of biopsies at BA46. A** Formalin-fixed tissue from three human brains was selected from the brain collection at Aarhus University Hospital. The red box marks the excised area of the tissue. **B** Fixed coronal block of tissue containing BA46. **C** Manual delineation of BA46 performed by MATLAB. **D** Infused image between global threshold image and sample area. The filtered surface of a coronal block of tissue containing BA46 was marked with a red and yellow map that shows the available sample area. **E** Summation of all white pixels for each row of the binary picture of the sample area. The 1st, 2nd, and 3rd quarter of pixels were marked with a green dashed line. **F** The two biopsies can only be sampled in either the 1st and 3rd quarter(white area) or the 2nd and 4th quarter (red area). In this case, random points in the 2nd and 4th quarters, marked with blue dots, were chosen by the algorithm. **G** The two chosen biopsies are marked with blue dots on the original block of tissue from (**B**).

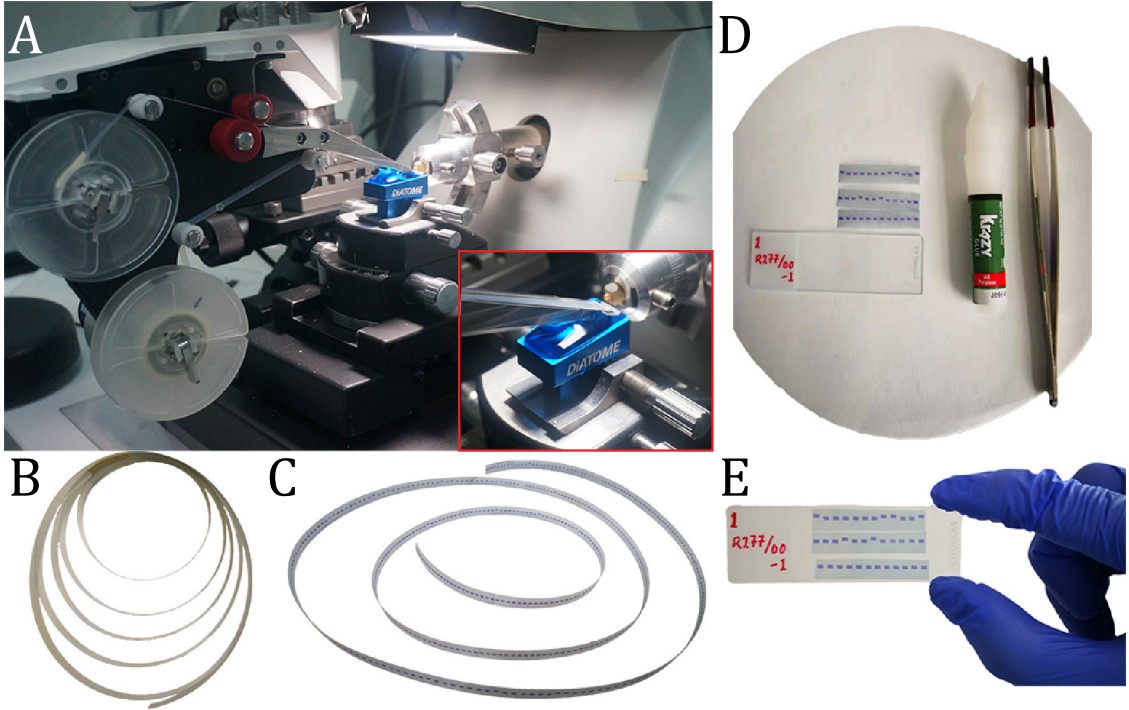

**Fig. 2 Image of the AutoCUTS-LM and sample preparation on tape. A** The supply-reel of the AutoCUTS-LM contains a transparent plastic tape that collects sections from the knife boat to the final take-up reel. (Red frame) Close-up view of the tape conveyor belt positioned in front of the knife boat with the mounted sample. **B** Collected sections on the transparent tape before staining. **C** Sections have been stained with toluidine blue. **D** Setup to glue the sections on the microscopic glass slide. **E** Stained sections glued on a microscopic slide and ready for image acquisition.

made the sections more adhesive to the tape. Thus, the sections would later stick to the tape during collection and staining.

Automated collection of the resin-embedded material was accomplished using a ultramicrotome attached to a custom-tape collection device (AutoCUTS-LM). We collected around 2400 serial sections with a total tissue depth of about 0.7 *mm* for each subject. The three human subjects' cutting thickness was chosen to be 400 *nm* for subject 1 and 300 *nm* for subjects 2 and 3

(sectional area was about 1.5 *mm*$^2$, and the total volume was ~1 *mm*$^3$). Sections were cut continuously with a diamond knife (Diatome, Switzerland) with an indoor humidity around 85%, see Fig. 2A. The pulling motion from the collecting tape moved the sections from the water onto the tape's surface, and its adhesiveness affected how flat these sections lay on the tape. Around 800 sections were collected per hour with our settings. Hence, we used less than 3 h to cut a tissue sample that is around

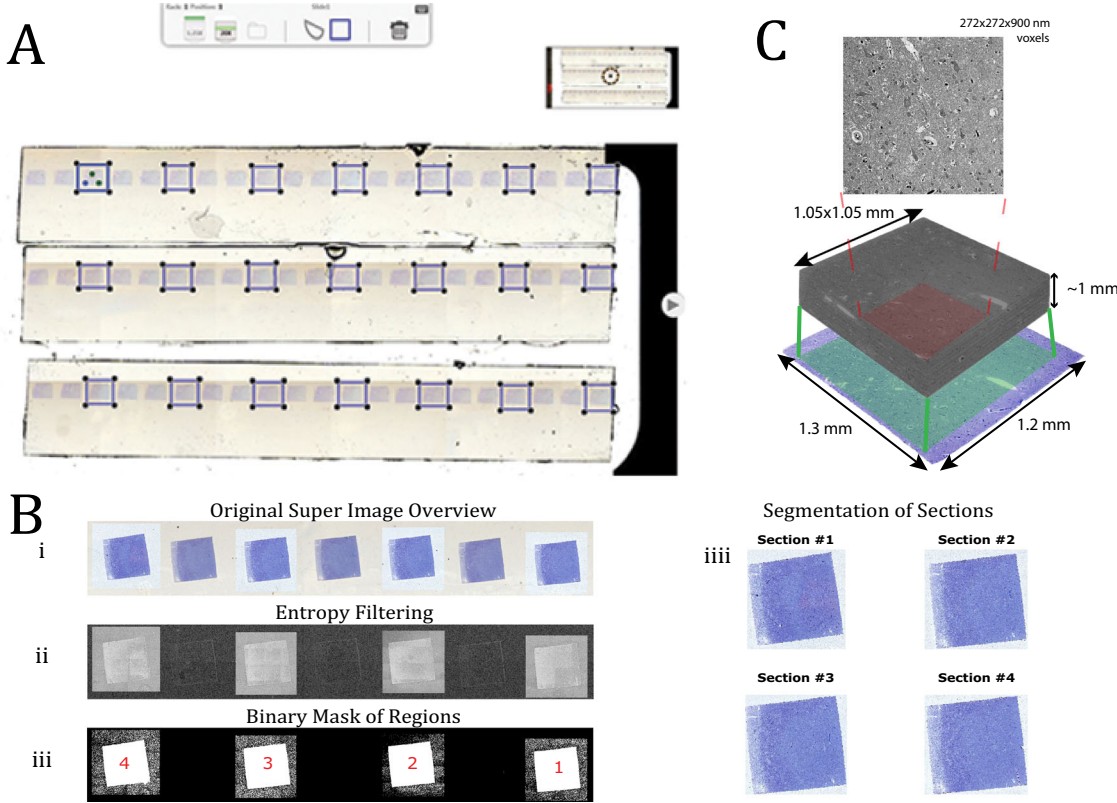

**Fig. 3 Image acquisition and 3D-stacks of aligned sections. A** Overview of a microscopic glass slide with the commercial Leica software. Systematic sampling of every third section was manually marked with three local points for the autofocus calibration. **B** In MATLAB, a TIF image was loaded where each second was of interest, which is different from (**A**) but easier to visualize as an example (i). Next, we used an entropy filter to detect sections of interest (ii). Binary masks have been computed for each section of interest and the images are ready for export (iii). The output of each exported section (iv). **C** Individual sections were aligned and stacked on top of each other. The stacked block of sections was then cropped down to a specific ROI (1.05 × 1.05 mm) which only contains tissue.

0.7 mm thick. It is important to notice that the ultramicrotome calibrates itself after sectioning the maximum useful range of 200 μm and had to be manually reset. In our case, a tissue sample to a depth of 0.7 mm could be sectioned with only three interruptions, and it was possible to move the sample to a different knife-edge, as dulling of the knife affects the section quality. Collecting thousands of sections was possible without any loss of tissue. However, we observed different technical and environmental factors that could generate wrinkles and disfigure the sections during cutting. The indoor humidity level was one of the main factors (see the "Methods" section).

Following section-collection, we chose to stain our sections with toluidine blue, since it interacts with most cells in the brain (both neurons and glial cells) and is thus excellent for revealing the neuronal patterns. However, to assess other biological results, methods such as immunolabeling or EM may also be used with the same AutoCUTS strategy, see Fig. S3. The spools holding the tape with attached sections were dried overnight in an oven at 50 °C. Sections were then stained with toluidine blue, and the tape with attached sections was cut into three consecutive strips with ~60 serial sections and glued onto a standard microscope slide (Fig. 2B–E). Each sample resulted in a section library of ~40 glass slides, which were digitalized.

**Data acquisition**. Digital images were acquired using the Apiro Versa 200 platform from Leica. The scanning speed was ~15 min per glass slide and required about 10 h to complete a section library of glass slides for each subject. We only sampled every

second and third section because a distance of 800 nm (subject 1) and 900 nm (subject 2 and 3) between sections was considered sufficient for 3D-reconstruction of pyramidal cells, which have an average somal diameter of ~13 μm[34], see Fig. 3A. The microscope included the commercial software Aperio ImageScope (Leica Biosystems Imaging, Inc., USA), which visualized the whole microscope slide image with high resolution. However, the user had to extract a region manually before an image could be exported as a TIF file. The uncompressed images were then automatically processed, ordered, and exported as individual images of each section using a MATLAB algorithm we developed, as shown in Fig. 3B. These individual images were then aligned and stacked by sequential image registration using our custom MATLAB scripts[35]. Next, the stack of images was prepared for further analysis by cropping tissue regions, see Fig. 3C.

**Pixelwise performance of deep learning model for segmenting pyramidal cells**. We worked with UNetDense, a deep learning framework, which has provided promising results for image segmentation of pyramidal cells[36]. The performance of the UNetDense model was measured by reporting sensitivity, precision, and F1-score. Neither recall nor accuracy was calculated as the cell's pixel count is dominated by background pixels making these measurements less informative. These measures were calculated based on the confusion matrices shown in Table 1. In general, the output of using individual models per subject showed a better predictive result compared to the combined model, see

**Table 1 Pixelwise validation.**

| Model | | 1 (A) | | | 0 (A) | | |
|---|---|---|---|---|---|---|---|
| | | Subject 1 | Subject 2 | Subject 3 | Subject 1 | Subject 2 | Subject 3 |
| Individual | 1 (P) | 128937 TP | 83370 TP | 91266 TP | 11710 FP | 14400 FP | 10717 FP |
| Models | 0 (P) | 19923 FN | 17810 FN | 11512 FN | 4033734 TN | 4078724 TN | 4080809 TN |
| Combined | 1 (P) | 119351 TP | 77409 TP | 92189 TP | 9506 FP | 12994 FP | 12337 FP |
| Model | 0 (P) | 29509 FN | 20717 FN | 8931 FN | 4035938 TN | 4083184 TN | 4080847 TN |

Pixel to pixel comparison between an MS and UDP image (2048 × 2048) for each subject. The confusion matrix is used to validate the UNetDense model on images that the model has not encountered before. The result shows the performance of individual trained UNetDense models for each subject and a combined UNetDense model used for all subjects. Pixels belonging to the background have label 0, those belonging to the pyramidal cells have label 1, (A) stands for actual (value in MS image), and (P) stands for predicted (value in UDP image).

**Table 2 Performance of pixelwise validation.**

| Model | Metrics | Subject 1 | Subject 2 | Subject 3 |
|---|---|---|---|---|
| Individual Models | Sensitivity | 0.87 | 0.82 | 0.89 |
| | Precision | 0.92 | 0.85 | 0.89 |
| | F1 | 0.89 | 0.84 | 0.89 |
| Combined Model | Sensitivity | 0.80 | 0.79 | 0.91 |
| | Precision | 0.93 | 0.86 | 0.88 |
| | F1 | 0.86 | 0.82 | 0.90 |

Precision, sensitivity, and F1-score of 2048 × 2048 pixel image for each subject with individual trained UNetDense models and one combined UNetDense model. Pixel evaluation is based on the confusion matrix of Table 1.

**Table 3 Performance of objectwise validation.**

| 3D-stack | TP | FN | FP | Sensitivity | Precision | F1 |
|---|---|---|---|---|---|---|
| Original | 472 | 19 | 33 | 0.96 | 0.93 | 0.95 |
| Filtered | 368 | 6 | 29 | 0.98 | 0.93 | 0.95 |

Validation of segmentation of pyramidal cells based on 3D-reconstruction. Precision, sensitivity, and F1-scores were calculated based on 491 reconstructed pyramidal cells from 30 stacked 2048 × 3840 manually marked images. Original refers to the case where centroid points from all 30 images are included. Filtered refers to the case where centroid points from the first and last three images of the stack were excluded.

Table 2. In general, individual models for each subject performed well with sensitivity, accuracy, and F1-score above 0.8.

**Objectwise performance of 3D-reconstruction of segmented pyramidal cells**. The performance of detecting pyramidal cells as 3D objects was evaluated by measuring the sensitivity, precision, and F1-score based on 3D-reconstructions of cells from a stack of 30 images. Manually Segmented (MS) and UNetDense Predicted (UDP) 3D-reconstructions were compared by checking whether estimated centroids from the MS cells fell within a cell profile of the UDP cells and vice versa. The sensitivity, precision, and F1-score were 0.98, 0.93, and 0.95, respectively, after removing datapoints from first and last three images, see Table 3. Thus, the 3D-reconstruction of pyramidal cells demonstrated a high performance across all three measurements.

**3D-reconstruction and morphological analysis of pyramidal cells**. Layer III was located using a density plot of a 2D-projection of the centroids of every 3D-reconstructed neuron, with yellow representing areas of high density and blue representing areas of low density, see Fig. 4. Layer I got a very low density of neurons, while II and IV are denser than layer III in BA46. As Layer III has a smaller density than Layer II and Layer IV, the ROI is specified between the two dense yellow areas for our analysis.

Further classifications into pyramidal or non-pyramidal neurons were needed as the UDP detects all neuronal shapes from 2D images. This is necessary because pyramidal cells' top

and bottom image profiles can be mistaken for smaller neurons/ glial cells, as seen in Fig. S4. The Gaussian mixture model (GMM) was used to classify the 3D-reconstructed pyramidal and non-pyramidal cells based on estimated sphericity and volume, see Fig. 5B. Big objects/cells were classified as outliers if the maximum Feret diameter in 2D or 3D measurements was three standard deviations from the mean. As a result, a total of 1, 19, and 28 datapoints for each subject were deemed outliers and hence excluded from the pyramidal cell data. The percentage of objects/neurons excluded from analysis using GMM and outlier detection accounted for 23, 25, 37, and 23% of the total number of detected objects for each dataset, as shown in Tab. S1. The mean density of pyramidal cells in layer III of BA46 after filtering was $28,155\ mm^{-3}$, and the GMM categorization and data containing outliers are shown in Figs. S5–S6. The measurements and calculations for each subject's entire stack were examined just for the classified pyramidal cells, with the number of cells and the size of the ROI window for each subject shown in Table S1. Table 4 provides information on the size, shape, and orientation of pyramidal cells of layer III in BA46 for all three subjects.

The average neuronal volume across all three subjects is $795\ \mu m^3$, and the shapes of pyramidal cells were assessed by approximating sphericity, giving an average value of 0.35[37]. The orientations of pyramidal cells relative to the direction of the vector pointing toward the pial surface had an average of 29°, and some examples of orientation vectors are shown in Fig. 5E.

The diameter was calculated using the nucleator probe by measuring the segment length from the largest cell profile ($Dia_L$) and the average segment length from all cell profiles ($Dia_{All}$). The average neuronal diameter for $Dia_L$ and $Dia_{All}$ were 11.17 and $7.03\ \mu m$, respectively. Estimated spheres of length $Dia_L$ were constructed and displayed with their matching 3D-reconstructed cell in Fig. 5F. Histograms of the different measurements can be found in Fig. S7 and as log-normal transformed in Fig. S8.

**2D vs 3D comparison of pyramidal cells sizes**. The exact same neurons were used to compare the 2D and 3D analyses for each subject. The volumes of pyramidal cells from 2D images were approximated by constructing a spherical object based on the estimated radii measured by the nucleator probe. The volumes of cells from the three subjects were calculated from 2D images using the estimated segment length from the largest cell profile ($Vol_L$) and all cell profiles ($Vol_{All}$). The average neuronal volume in 3D ($Vol_{3D}$), $Vol_L$, and $Vol_{All}$ are 795, 730, and $183\ \mu m^3$, respectively.

An approximation to the mean diameter of a cell $Dia_{3D}$ was estimated from the mean volume $Vol_{3D}$ under the assumption that it is the volume of a perfect sphere. $Dia_{3D}$ was then assessed and compared to the diameter measured from the largest cell profile ($Dia_L$) and all cell profiles ($Dia_{All}$). Because the nucleator probe is derived from the mathematical fact that the length of isotropic test lines between a unique point and the cell border, it

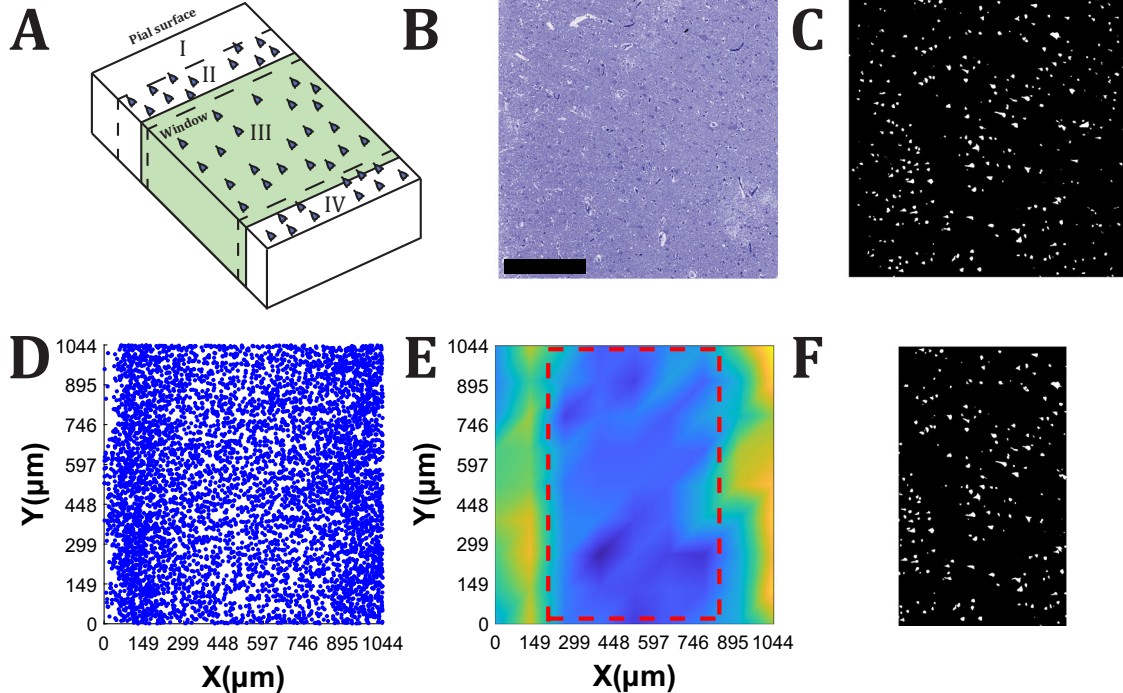

**Fig. 4 Visualizing the steps to identify layer III. A)** Illustration of four layers of BA46 and the observation window for the analysis is marked in green. **B** Raw image of a section. Scale bar = 300 μm. **C** Binary image of the output from the Deep learning model, UNetDense. **D** Position of centroids from 200 images projected to the *x*- and *y*-plane. **E** Density map of the positions of neurons to visualize the different layers in the cerebral cortex and the yellow color represents areas of high density, whereas the blue color represents areas of low density. The density was high at the yellow areas, which indicate the position of layer II and layer IV (from left to right) and low at the blue areas, which indicate fewer neurons and show a part of layer I and layer III. The volumetric stack of images was cropped within the squared marked with the red dashed lines that was selected by a user. **F** The part of the image in C, which is chosen for analysis.

provides the most accurate one-to-one comparison. The average diameter for $Dia_{3D}$, $Dia_L$, and $Dia_{All}$ are 11.48, 11.17, and 7.03 $\mu m^3$, respectively. The values for the 2D vs 3D comparison of volume and diameter can be seen in Tab. S2–S3.

**Point pattern analysis of pyramidal cells**. The coordinates of the centroids for the 3D analyzed pyramidal cells in layer III of BA46 form a spatial point pattern. We considered four such point patterns, which we refer to as 1_1, 1_2, 2, and 3 and they correspond to the three subjects (Subject 1 was divided into two parts since it was collected over two different days).

In order to detect possible columnar structures in the datasets, we estimated the cylindrical *K*-function for each dataset, and compared it to the 95% global envelope obtained by simulations under the null hypothesis of complete spatial randomness (CSR). The results can be seen in Fig. 6. We considered the empirical cylindrical *K*-function in the directions of the three main axes. When the empirical cylindrical *K*-function is above the envelope, it indicates cylindrical clusters of points (centroids of cell locations) in the corresponding direction.

There were signs of columnar clusters in all three directions for all subjects. However, it was most pronounced in the direction of the *x*-axis, which is the expected direction of the possible columnar structure of pyramidal cells, especially when looking at radii between 5 and 20 $\mu m$ and heights between 20 and 80 $\mu m$. There were areas where the empirical curves were below the envelopes suggesting some repulsive behavior in the data. This was not unexpected since the point patterns only represent the centroids of cells, since cells cannot overlap, it was thus natural to see some repulsion between the points. The global envelope tests corresponding to the situations in Fig. 6A all yielded *p*-values

below 0.05, and the tests corresponding to the situations in Fig. 6B all yielded *p*-values below 0.001, indicating that all datasets exhibited large deviations from CSR.

**Tissue deformation**. Every single-cell location research technique is vulnerable to tissue deformation. Epon embedding is used for the analysis in this paper to minimize shrinkage. We compared the tissue area of three Epon-embedded biopsies before and after processing. The human gray matter brain biopsies showed an areal shrinkage equal to 0.1%, 4.2%, and 7.9%, respectively. Based on these data and our previous publications[38,39], it was decided not to correct our results for shrinkage.

## Discussion

The presented method can reveal fundamental characteristics of specific brain areas, which can be used to improve our understanding of neuronal morphology and their spatial relationship. A variety of efforts have been made toward developing technologies for revealing the intricate patterns of neural circuits.

By using different slicing or optical methods to recreate neuronal tissues, multiple studies have provided 3D data on the microscopic morphology or gene expression of neurons[6,40–46]. Due to a number of difficulties, such as size limitations of the slide scanner and microtome, low homogeneity of serial whole organ sectioning and staining, the time-consuming design of the operation, computer constraints, and limited digital processing capability, the entire human brain is difficult to recreate entirely on a mesoscopic scale. High-resolution, 3D models of animal and human brains are difficult to acquire. Chemical clearance methods make the tissue transparent and enable the entire mouse brain to be reconstructed[6,44], but 3D-reconstruction and

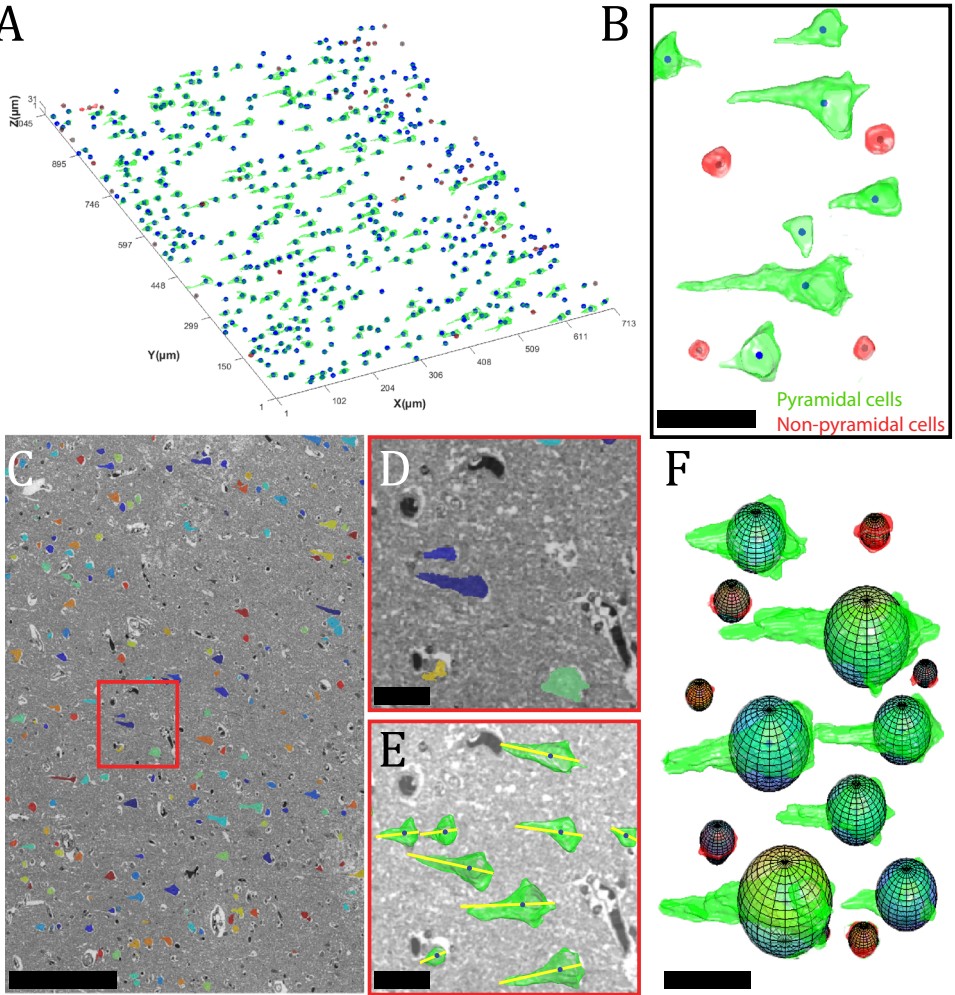

**Fig. 5 Illustration of 3D-reconstruction of neurons and segmentation of pyramidal cells. A** Visualization of the 3D-reconstruction of one layer neurons with a window height of 31 μm. **B** Zoomed image of the segmented pyramidal and non-pyramidal cells. Scale bar = 20 μm. **C** Overlay image of the binary segmented image with the gray-scale image of a section. Scale bar = 200 μm. **D, E** Close-up view of the overlay picture and 3D-reconstruction of pyramidal cells, with yellow lines representing orientations. Scale bar = 15 μm. **F** Measurement of diameter from the longest cell profile $Dia_L$ was utilized to construct 3D spherical objects with their corresponding 3D-reconstructed cell. The graphic indicates that the difference in 3D volume space between the spherical approximation and the 3D-reconstructed item is similar. Scale bar = 20 μm.

**Table 4 Quantitative values for pyramidal neurons from the entire stack of each subject.**

| Brain | Volume (μm³) | Orientation (°) | Sphericity | $Dia_L$ (μm) | $Dia_{All}$ (μm) |
|---|---|---|---|---|---|
| Subject 1 | 867 ± 960 | 34 ± 23 | 0.38 ± 0.08 | 11.13 ± 3.46 | 7.23 ± 3.26 |
| Subject 2 | 709 ± 806 | 28 ± 22 | 0.33 ± 0.07 | 11.13 ± 3.74 | 6.49 ± 3.1 |
| Subject 3 | 808 ± 893 | 24 ± 18 | 0.35 ± 0.06 | 11.27 ± 3.61 | 7.37 ± 3.1 |
| Mean | 795 ± 65 | 29 ± 4.1 | 0.35 ± 0.02 | 11.17 ± 0.07 | 7.03 ± 0.39 |
| CV | 0.08 | 0.14 | 0.06 | 0.01 | 0.06 |

The table shows the neuronal volume, diameter, orientation, and sphericity of pyramidal cells for each subject based on the data which is summarized in Fig. S7. The entries for each subject state the average and ±one standard deviation. Mean is the average measurement for each column, coefficient of variation CV = SD/mean.

digitization of fine neuronal morphology continue to be challenging. One explanation is that in the nervous system, cells are closely arranged, making it difficult to distinguish one from another. Applying Golgi-staining, a 3D structural dataset of the entire mouse brain was also collected by micro-optical sectioning tomography, which can conduct imaging and sectioning simultaneously on centimeter-sized tissues[42]. However, such a section-based method demands costly specialized instruments, relatively long periods of sample treatment (weeks or months), and for a single brain imaging time can approach even one month[47]. Based on the reconstruction of histological sections of a human brain preserved in paraffin, Amunts et al. developed a 3D model of the whole human brain called BigBrain with a spatial resolution of 20 μm[45]. The tissue was embedded in paraffin, which can shrink tissue volume up to 50–60%[39], and it is challenging to study neuronal morphology with a spatial resolution of 20 μm. To date, no optimal approach to the analysis and processing of the whole human brain with micro-resolution has been achieved. The

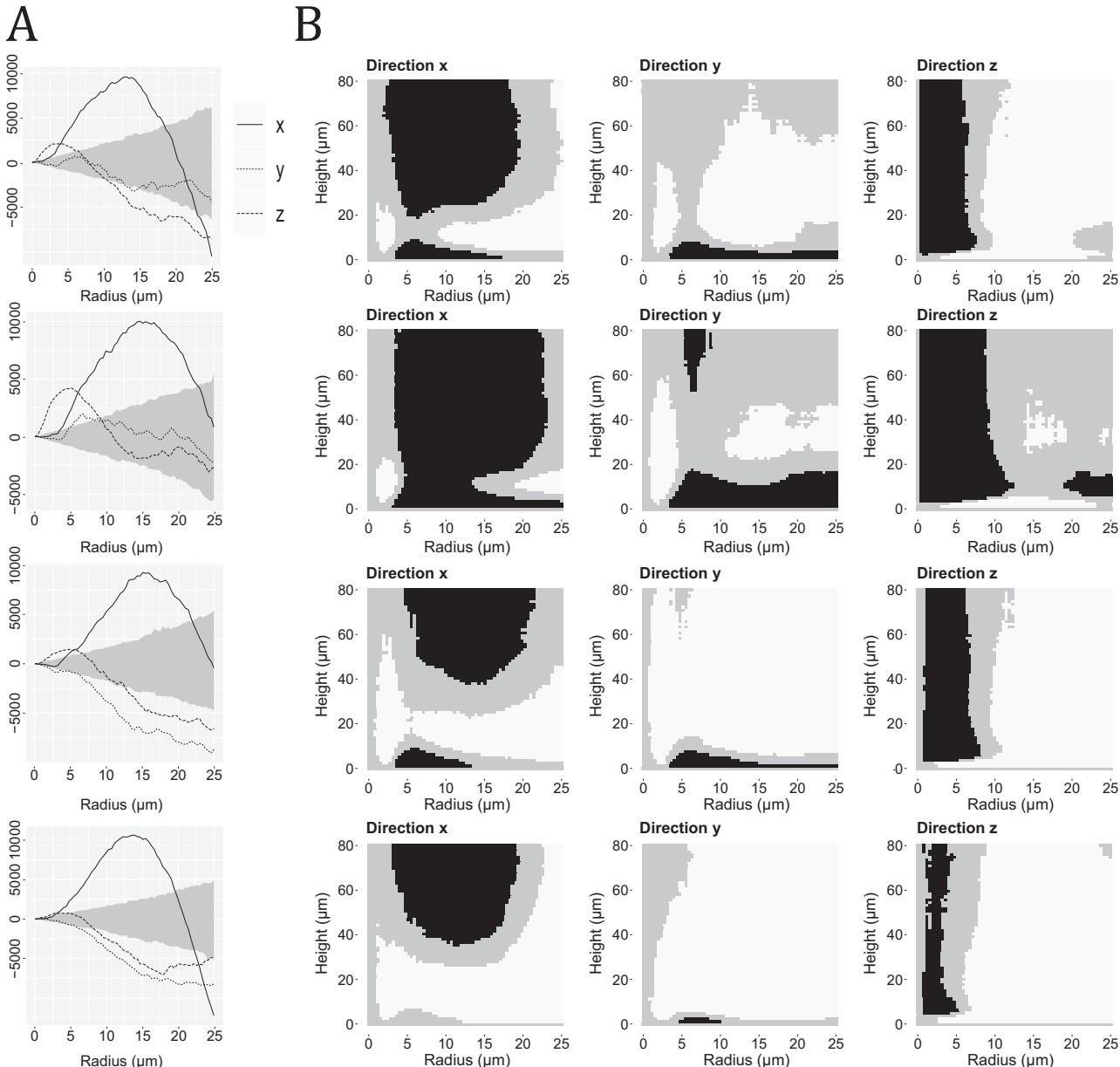

**Fig. 6 Results of analyses with the cylindrical *K*-function for the spatial point patterns of centroids of pyramidal cells.** Each row represents a dataset which from top to bottom are Subject 1_1, 1_2, 2, and 3. **A** 95% global envelopes (shaded area) for the cylindrical *K*-function, with $t = 80$ fixed, and using 2000 CSR simulations. The theoretical value of the cylindrical *K*-function under CSR was subtracted from the curves for better visualization. The three curves correspond to the empirical cylindrical *K*-function for each dataset, in the direction of the *x*-axis (solid lines), *y*-axis (dotted lines), or *z*-axis (dashed lines). **B** Summary of 95% global envelopes for the four datasets based on the cylindrical *K*-function when varying both the height *t* and radius *r*. The envelopes are each based on 4000 CSR simulations. The direction of the cylinders is stated at the bottom of each plot. The plots indicate whether the empirical cylindrical *K*-function for the observed point pattern is above the envelope (black), within the envelope (gray), or below the envelope (white).

majority of the approaches discussed above share the use of immunolabeling in order to detect and localize antigens or proteins within a cell at a specific location.

The fundamental constraint of immunolabeling is that milder fixation conditions and a shorter detergent permeabilization incubation time are required to allow antibodies to penetrate the tissue. This is particularly important for soluble proteins in the cytoplasm, which are frequently damaged or destroyed during incubation. Hence, it is challenging to create neuronal antibodies that work in postmortem brains that have been in the fixative for months or years[48]. Another well-known concern is that many histological procedures can cause tissue samples to shrink and deform, potentially altering cell size, shape, and organization.

Keeping the dimensions of the tissue is especially essential if studies want to assess changes in the size and distances of any cells or organelles within the tissue. Our methodology provides an effective technique for imaging smaller pieces of most archival tissue from semi-thin serial sections into a functional dataset with hardly any tissue deformation. Researchers have with our method a unique opportunity to study archived tissue samples, enabling researchers to investigate tissue and disease development in great detail.

An automated section collection EM system has been altered into a unique 3D-reconstruction method for light microscopy, AutoCUTS-LM. We used histological staining to visualize cytoarchitecture and an autonomous slide scanner to produce

high-resolution images of the brain tissue. Via deep learning, pyramidal neurons were characterized and their spatial distribution analyzed using advanced techniques. Our application provides valuable information on the neuronal 3D architecture from archived human brain tissue.

We systematically sampled the tissue for our setup to reduce the scanning time and the processing of data. A thickness of 800 and 900 $nm$ provided sufficient axial spatial resolution to detect and reconstruct pyramidal cells. The UNetDense architecture was applied to detect the pyramidal cells in our images, as threshold-based segmentation techniques typically yield poor output in medical image analysis for low-contrast images, unexplained noise, blurred boundaries, and different light condition[49,50]. The variation between brains made it challenging to create a model that suited all three brains since flaws occurred mainly along the edges of the images, and where the contrast was low due to less staining absorption. Therefore, we trained an individual UNet-Dense model for each subject in this study since the sensitivity, precision, and F1-scores for both pixelwise and objectwise segmentation were above 0.82 and 0.93, respectively. Overall, the results of the combined model showed smilar precision and F1-scores but lower sensitivity compared to the individual models. That being said, sensitivity was emphasized for this study as it demonstrates how effectively the model detects pyramidal cells present in the data, which is essential for conducting the point pattern analysis. The risk of having three models is overfitting. Overall, the validation-findings from the sensitivity, precision, and F1-scores showed that the outcomes of the three models are reliable. One reason for the difference between subjects was that the brains absorb staining differently, and this may be due to different ages, postmortem intervals, and long fixation time. With this limitation in mind, one model might be adequate if the captured images did not alter too much or if we annotated more images to train the model.

We measured the volume, sphericity, orientation, and diameter of pyramidal cells and used the cylindrical $K$-function to identify columnar structures. The average volume was 795 $\mu m^3$ with an equivalent diameter equal to 11.48 $\mu m$. The neuronal diameters of pyramidal neurons in layer III of BA46 have received little attention in the literature. Nonetheless, Rajkowska et al. examined 150–200 neurons in this region using stereological techniques and determined an average diameter of 13.45 $\mu m$. The process for measuring neuron radii was comparable to the nucleator probe as they measured the border outlines of cell profiles to compute the associated diameter. Similar to the procedure in this study, they only assessed neurons in the crown of a gyrus. The differences in radii could be related to the fact that we sampled pyramidal cells from a small concentrated area. Also, they quantified neurons in broader regions of BA46 using a succession of counting boxes. In comparison, our method offers information on thousands of counted neurons from each subject, whereas Rajkowska counted 150–200 neurons per subject[34]. In BA46, counting fewer neurons from smaller sample sizes over a broader area may overestimate neuronal radii. This is due to the fact that smaller pyramidal cells dominate layer III over bigger ones, resulting in a stronger proclivity toward smaller radii. The difference in radii measurements across individuals might also be related to biological variance for subjects, brain storage period, or different embedding media, as the difference is roughly 17%.

Different factors such as age, disease, and toxicity have been reported to affect neuron volume in the cerebral cortex across other studies, however, they do not give any detail on how the shape of the neurons are affected[51–54]. A literature search revealed that no quantitative values to quantify the sphericity and orientation of pyramidal cells had previously been published. As a result, such measurements were not comparable to those of other studies.

Sphericity is valuable as a general shape descriptor since it also applies to objects having holes, such as a torus. Yet, the main reasons sphericity is used for measuring shape are as follows: sphericity is a unitless number, hence is scale invariant. Furthermore, it is simple to compute since MATLAB easily calculates the surface area and volume of cells, which are fundamental descriptors of cells in their own right. Estimation of sphericity does not presume any shape before examining a 3D object, making it an appealing tool for differentiating between distinct forms. As a result, we implement sphericity estimates to comprehend a natural cell shape. The average sphericity of 0.35 found in this paper indicates that pyramidal cells appear elongated and far from spherical.

The average angle between the vector that represents the orientation of the pyramid cell and the $x$-axis, which points to the pial surface, is 29°, suggesting that pyramidal cells are focused perpendicular to the pial surface. Quantities such as cell sphericity and orientation in follow-up studies with different disease groups could be utilized to understand the morphological alterations induced by these disorders.

The volumes of pyramidal cells calculated from 3D-reconstruction images were compared to the results of the nucleator probe measurements for the same pyramidal cells, which estimate the mean cell volume from 2D images. It is crucial to keep in mind that the volume estimates are based on two separate methodologies, and the volume may be overestimated or underestimated with either approach. The 3D approach measures every cell profile, and if a cell profile on the edge of a pyramidal cell is not recognized, this approach may underestimate the volume (top or bottom cell profile). On the other hand, the nucleator relies on measurements from a unique position (e.g., nucleolus), assuming that the section or cell is isotropic. If this criterion is not fulfilled, the estimate may be biased.

The average estimated volumes of pyramidal cells were 795, 730, and 183 $\mu m^3$ for $Vol_{3D}$, $Vol_L$, and $Vol_{All}$, respectively. The difference between $Vol_{3D}$ and $Vol_L$ is 8.7%, whereas the difference between $Vol_{3D}$ and $Vol_{All}$ is 77%. It appears that using the nucleator probe on the largest cell profile to estimate volume yields a comparable overall result, but employing the nucleator probe on all cell profiles causes a huge difference. The estimated average volumes vary greatly depending on the approach used.

The estimated diameter of pyramidal cells changed less than the volume estimates, with results of 11.48. 11.17, and 7.03 $\mu m$ for $Dia_{3D}$, $Dia_L$, and $Dia_{All}$, respectively. The difference between $Dia_{3D}$ and $Dia_L$ is 2.9%, whereas the difference between $Dia_{3D}$ and $Dia_{All}$ is 39%. Hence, if we look at estimated radii when employing the largest cell profile compared to the 3D estimation, there is no substantial disagreement between those two approaches for calculating diameter because the difference is nearly equivalent to the 272 $nm$ pixel size. In contrast, estimating the average radius from all cell profiles makes a notable difference. As the neurons are not spherical in reality, comparing neuronal size based on volume rather than diameter is a more appropriate parameter to quantify changes in neuronal size in future references. Nevertheless, if only a 2D technique is available, adopting a spherical approximation by assuming general isotropy of these human pyramidal cells to estimate neuronal volume is not entirely inaccurate. It is vital to notice that a slight change in the radius can substantially change the volume value considering the radius is raised to the power of three while computing the volume.

Even though the nucleator is a 2D procedure, it is nevertheless based on a 3D sampling methodology since the biggest cell profile detected during the sample is required before the analysis can be completed. The nucleator is therefore a helpful tool to use when it is challenging to distinguish smaller cell profiles at the edges and

only larger cell profiles are required. Another important consideration is that the nucleator will be less sensitive to tissue shrinkage in the direction of the $z$-axis as it estimates the volume from a single 2D plane. The nucleator is advantageous for estimating volumes compared to other embedding materials that are sensitive to shrinkage, such as frozen sections or vibratome sections, which rarely deform in the $x$- and $y$-axes but shrink in the $z$-axis.

The observed pyramidal neuron density in layer III of BA46 after the classification between pyramidal and non-pyramidal cells is 28,160 $mm^{-3}$. A stereology study by Francine M. Benes found a neuronal density of 36,800 $mm^{-3}$ in layer III of postmortem brains from nine healthy controls in the prefrontal cortex[55]. Cullen et al. estimated a neuronal density of 37,000 $mm^{-3}$ in the prefrontal cortex of 10 adults with no history of mental illnesses, similar to Benes et al. However, Cullen et al. also measured the density of pyramidal cells in layer III to be 25,650 $mm^{-3}$, a difference of 30% compared to the total neuronal density. Rajkowska et al. characterized and mapped the cytoarchitecture of BA46 in 17 healthy people. They were able to estimate a neuronal density of 51,510 $mm^{-3}$ by blending cortical layers I–III[34]. Because the density in layer II varies from 48,000 to 78,000 $mm^{-3}$ and covers a smaller area than layer III, it is reasonable to assume that the density in layer III would be about 36,000–44,000 $mm^{-3}$ [55,56]. If the difference between neuronal and pyramidal density is roughly 30%, then the projected density should be ~25,200–30,800 $mm^{-3}$, which is consistent with our findings.

One thing all of the three studies listed above have in common is that they solely measured prefrontal cortex neuronal density[34,55,56]. Number densities indicate changes in the number of cells as well as tissue volume under consideration, because they are ratios. It is commonly assumed that as the number of detected objects rises, so does the density. However, the density increases as well if the number of identified objects remains constant, but the examined volume shrinks. As noted in other studies, making any definite claim regarding changes in 3D structures based purely on density estimates may lead to questionable findings[57–59]. Because this assumption is so widespread, the phrase "reference trap" refers to circumstances in which wrong conclusions are drawn only on the basis of density. As a result, comparing densities should be handled very carefully.

Some studies have shown that pyramidal cells are organized in columns that are perpendicular to the pial surface of the cerebral cortex[60–63], other studies suggest otherwise[64,65]. It is an attractive idea to explain the neuronal organization because interconnected neuron groups typically share similar physiological properties, and the conditions that excite a neuron are also likely to excite a considerable fraction of its afferent input. The loss or changes in the spatial organization of neurons may interfere with information processing between distributed networks, thereby promoting cognitive decline. The spatial distribution of pyramidal neurons was analyzed with the use of the cylindrical $K$-function, which does not assume any columnarity a priori, and where we applied the cylindrical $K$-function on much larger point pattern datasets than so far analyzed in the literature[66–68]. Our results suggest that there is evidence of a columnar structure in the directions of the $x$-, $y$-, and $z$-axes for all three subjects, but it was most pronounced in the direction of the $x$-axis, which is pointing toward the pial surface. Thus, the results support the theory of a columnar pattern perpendicular to the pial surface. The method can be used to detect potential cytoarchitectural distortions that may impact the neuronal columnar organization in the brain cortex. However, more human brains need to be studied in order to draw any final conclusions.

Technical and environmental factors could affect the sampling quality of our sections. The plasma treatment required the transparent collection tape to be hydrophilic to ensure that the sections would adhere to the tape[20,69]. If the tape was too hydrophilic, the sections would not have time to unfold before they landed on the tape and folds were unavoidable. Variations in density between tissue samples and resin meant that the remaining blank resin had to be trimmed off[4,20,70]. We observed the advantage of higher indoor humidity, which provided a favorable environment for collecting sections. The impact of induced folds can be seen with a humidity level about 10%, 60%, and 90% in Figs. S10–S11. The test showed that a stable indoor humidity level around 80% and 90% could help avoid folds.

If larger structures were to be explored in the future, the use of an optimized resin embedding protocol that could be used for thicker sections (>500 $nm$) would ease the workload. Applying a resin ratio test to match the tissue density and select the most suitable one before collecting samples with AutoCUTS-LM would have a beneficial effect on reducing the folds in the section. Reducing the hardness of the resin to suit a particular tissue type can be achieved by controlling the ratio of two different anhydride curing agents (Dodecenyl Succinic Anhydride and NADIC Methyl Anhydride). This could avoid the excessive cutting of thousands of needless sections, which would result in the sharpness of the knife being extended, using less tape, and a reduced workload for data collection and image processing. Other labeling methods can also be carried out with the AutoCUTS-LM, as our sections can be combined with immunofluorescence labeling to identify the distribution and co-localization of proteins as shown in Fig. S3A, and it may also be combined with in situ hybridization in order to localize a specific DNA or RNA sequence[71]. Due to the use of a hard resin, sections can also be viewed in an electron microscope, Fig. S3B, in which case cellular ultrastructure can be visualized. In essence, this methodology can be designed for both light microscopy and electron microscopy to bridge the localization of important molecules with cellular ultrastructure.

In conclusion, the AutoCUTS-LM method could benefit research in cell and developmental biology, model organism analysis to address mechanisms as cell migration, 3D tissue modeling, and morphological changes between animal/patient groups. This method is applicable for any disease and can potentially enhance the research of normal and disease processes, particularly those involving morphological alterations or in which the spatial interaction of disease features is essential.

## Methods

**Subjects**. Three healthy human brains (two women, one male) aged 30, 53, and 58 (Subjects 1, 2, and 3) with no history of psychiatric or neurological diseases were selected from the brain collection at Core Centre for Molecular Morphology, Section for Stereology and Microscopy, Aarhus University Hospital, Aarhus, Denmark. These archived brains have been stored for 19, 21, and 21 years, respectively, in 4% formaldehyde in phosphate buffer at neutral pH and were collected in compliance with Danish law and with approval from the Central Denmark Area Health Research Ethics Committees (license number: M-2017-91-17).

**Sample extraction**. We have developed an algorithm in MATLAB that can assist any user in delineating a ROI in a block of tissue and systematically sample two (or more) points for biopsies on the cortex, see Fig. 1. The user had to capture a picture of the tissue block and define the biopsy diameter. Then the image was transformed into gray scale, and the ROI was delineated to produce the sample area. This picture was then transformed into a binary image indicating the sample area by means of a global threshold. Next, to prevent the biopsy from being placed on the crown of the gyrus, the edges of the binary image were removed. This was done by using morphological erosion with a filter size equivalent to the biopsy diameter and is shown with the red color in Fig. 1D. The sampling area was divided into four quarters. The two biopsy punches should either be performed in the first and third quarters or in the second and fourth quarters to avoid overlapping biopsies. The four quarters were determined by considering the accumulation of white pixels by rows and detect when they reached 25, 50, and 75%, see Fig. 1E. A 1.5 $mm$ diameter

biopsy punch was used to sample brain tissue covering all six layers of the cerebral cortex.

**Sample embedding and block preparation**. The biopsies were first immersed in Phosphate-buffered saline (pH 7.3) with sucrose for one day and then washed two times in 0.05 Mol maleate buffer (pH 5.2), 5 min each time, at room temperature. Osmium is traditionally used to stain samples for epoxy-resin embedding for EM. However, we discovered that it decreased the sample's signal-to-noise ratio (SNR) when it was applied for light microscopy, see Fig. S1. Samples were processed and embedded inside the Leica EM TP Automated Tissue Processor (Leica Microsystems, Brønshøj, Denmark). Here, they were stained with 1% uranyl acetate in maleate buffer for 1 h and dehydrated through a graded ethanol series (70% 86, 96, and 99%, 20 min each). Following the completion of dehydration, samples were washed three times with 100% acetone for 10 min, followed by infiltration in 100% acetone/epon 1:1 with constant rotation for 12 h overnight. Infiltrated samples were incubated in pure Resin 812 for 1 h and placed in embedding molds in a pre-warmed oven (60 °C) to polymerize for 24 h. The biopsies were placed in the bottom of the embedding form, such that the pial surface was perpendicular to the cutting direction of the knife, as sectioning all six layers in the neocortex was preferred. This reduced the number of sections by approximately two to three times compared to alternative orientations, which ease the time spend on capturing images for 3D-reconstruction. After the resin had fully cured, most of the white resin from the embedded sample was roughly trimmed by a high-speed milling system (EM TRIM2, Leica) with an angle set to 60°. A glass knife was used for fine adjustment to trim around a $1.1 \times 1.4\ mm$ rectangle with a depth of 0.7 mm ($1\ mm^3$), resulting in a sample where only neurons in the supragranular layers and layer IV were included, see Fig. S2. It was essential to trim all the blank epoxy-resin away since the density difference between the tissue and epoxy can generate wrinkles while cutting. Supplementary Note 1 describes in detail how to prevent wrinkles and optimize section quality, as seen in Figs. S9–S14.

**Transparent collection tape**. Collection tape with different settings was tested to find the most suitable tape for our needs. A roll of PEN tape 300-mm wide, 45-m long, and 50-μm thick was chosen for this study attributable to heat treatment and the feature of a protective coat on both sides, which prevents dirt (South China Science & Technology Co., Ltd, China). This tape was slit into 7-mm wide strips (Tianjian Xinhua Electronic Material Co. LTD., China). Adjusting the tape hydrophilicity was essential since it reduced wrinkles of the brain sections on the tape and made the sections more adhesive to the tape, so they did not fall off during collection and staining. The system parameters for the plasma treatment (Beijing Jiaruntongli Technology Co., Ltd., China) were set with the values: power 120 W, frequency 40 kHz, speed time 4 mm/s, and processing time of about 2 h for 20 m tape.

**Automatic serial section collection**. An ultramicrotome (EM UC7, Leica) connected to a custom-tape collection system (AutoCUTS) was used to automatically cut the resin-embedded material into serial sections. Serial sections with a thickness of 400 and 300 nm were cut by a 45° Histo diamond knife (Diatome, Switzerland) and floated onto the water surface, see Fig. 2A. The tape's reel speed and cutting speed were set to 1 and 2 mm/s, respectively, which gives a distance of 1 mm between every section on the tape. The pulling motion from the collection tape brings the sections from the water to the surface of the tape, and the adhesiveness of the tape affects the flatness of these sections.

We collected about 800 sections per hour with our current settings. Hence, it takes about 3 h to finish around 0.7 mm of tissue. A video camera was attached to the AutoCUTS to monitor and record the process to ensure a more comfortable experience for the user by displaying the cutting process onto a computer screen. It was possible to collect thousands of sections without any loss of tissue.

Different technical and environmental factors were observed that could generate wrinkles and disfigure the sections during cutting as described in Supplementary Note. We found that sections with a cutting thickness above 300 nm were more susceptible to generate these wrinkles during the cutting process. The impact of wrinkles was also affected by a combination of room humidity level and tape hydrophilicity.

**Section library**. The spool that contains the tape with attached sections had to be completely dry before staining was applied to prevent any sections from falling off during this process. Hence, the spool was placed in a sealed plastic bag and placed inside a 50 °C oven overnight. The toluidine blue staining was absorbed differently in each brain. Consequently, we had to check the optimal staining time for every brain before running the protocol. Some sections would be stained at room temperature for 20 min to decide which time the pyramidal cells had the best signal-noise ratio. Hereafter, the tape was segmented into smaller pieces and placed in a petri dish (20 cm diameter) filled with 1% toluidine blue without the tape sticking together Fig. 2B. During the staining period, a cover was placed on top of the petri dish to condense moisture and prevent dust or dirt from mixing with the blue toluidine solution. Toluidine blue residues on the tape segment were washed away (once with 80% ethanol, twice with purified water), and the tape segment was then dried with a hairdryer (Fig. 2C). It was important not to let the tape dry naturally

since water stains would then be developed on the plastic surface of the tape, which generates hazy white spots on the surface. The tape segment was cut into smaller strips and glued onto a typical 75 mm by 25 mm microscope slide (Fig. 2D, E). The sequence of the serial sections was numbered from the bottom right corner to the top left corner. Beforehand, each glass slide was cleaned of dust and other particles in pure acetone and alcohol.

Different glues were bought and tested for their ability to adhere PEN tape to the microscopy glass slide. For this study, we chose the glue from Krazy glue (Krazy Glue All Purpose Super Glue Pen, Fine Tip, 3 Grams) due to its high adhesive level and because it was non-toxic and convenient to use. Krazy glue does not require a fume hood, which can generate turbulence and blow away the cutting strips. Besides, the pen shape made it easier to use than a standard plastic pipette (Fig. 2D). Note that sometimes air bubbles could be generated between the glass and tape, so it was essential to press the tape as flat as possible with a pair of rubber-tipped tweezers. The 7-mm wide tape was chosen for this specific reason since it was more convenient to glue $3 \times 7\ mm$ parallel strips onto a 25 mm glass side compared to the traditionally 8-mm wide tape.

**Data acquisition**. Images of the sections were acquired using Leica's Apiro Versa 200 Digital Pathology Scanner. The Apiro Versa 200 is equipped with a 200-slide autoloader and a robotic arm that allows any user to capture pictures unsupervised.

First, it captured a low-resolution overview image of the whole slide. Subsequently, three focal points were positioned manually on each observable section for autofocus adjustment. Next, the microscope collected images at higher magnifications (lens ×20, NA 0.8) with pixel size down to 272 nm, which did not lose any details because the pixel size was below the expected resolution for this system's optics. Systematic sampling of our sections was done by only chosen every second (subject 1) and every third section (subjects 2 and 3). This choice was based on the chosen cutting thicknesses of 400 and 300 nm, which correspond to a sample interval of 800 and 900 nm, respectively.

The output files were named based on the positions of the glass slides in the loader, e.g., Slide1 for position 1, and the output files could be read from the commercial software Aperio ImageScope (Leica Biosystems Imaging, Inc., USA) that was part of the microscope interface. Aperio ImageScope could visualize the whole slide image and keep the high-resolution image. However, the user had to manually select a region before sections could be exported as individual image files. As a result, we built a script that could load large image files containing multiple sections and export the individual sampled sections as uncompressed TIF files in order, see Fig. 3B[72].

The output images for each glass slide were roughly 2–4 GB in total if images for each stripe were exported, and it is thus recommended to break the photographs into smaller segments for each glass slide. First, the large image files were converted to gray scale, followed by a Gaussian blur. An entropy filter was then applied to measure randomness, which was used to characterize the texture of the input image. This could be used to detect sections of interest since pixel values of sections in focus varied a lot due to the presence of various fine details, while sections out of focus were blurry and showed more homogenous pixel values. After that, a binary mask for each section was created by replacing all values above the globally defined threshold with 1 and filtering out smaller connected components. Each section detected was then exported as an individual file and prepared for alignment.

**Alignment of sections**. The alignment of the segmented sections for each subject was accomplished by a sequential slice-to-slice image-based registration approach. Regions in image pairs were matched by translating and rotating images following a precise registration. First, we converted the RGB image of a section into gray scale and then employed a median filter to remove the noise. Then, we did a rough registration followed by a fine-registration using rigid intensity-based image registrations with different optimizer and metric configuration properties[35]. In MATLAB, we set the imregconfig function, which determines the optimizer and the metric configuration, to the multimodal configuration, as images may have varying intensity distributions. The function imregconfig was set to its default values for the rough registration, while the growth factor, initial radius, and maximum iterations were updated to 1.02, $2 \times 10^{-3}$, and 300, respectively, for the fine-registration. For the image registration, images were downscaled by a factor of four in all directions before any transformations were done in order to increase the registration speed. After the transformation matrix calculation, the images were upscaled again in order to recover the original scales. After all the images were aligned, a window in which only the tissue remained was chosen. The images were chopped to this window and prepared for analysis, see Fig. 3C.

**Data analysis pipeline**. The research pipeline for processing microscope images is depicted in Fig. S15, where the key steps in the pipeline are explained in the following subsections. First, we manually annotated microscope images and augmented those images to produce a sufficient amount of images for the training and validation set used to train the UNetDense model. After all aligned images were segmented with the UNetDense model, a density map was used to identify multiple layers of neurons in the neocortex, which made it possible to identify and crop out layer III for further study. Then, we performed 3D-reconstruction and calculated

morphological parameters for all cells from the segmented neurons in layer III. On the basis of the 3D-reconstruction cells, pyramidal cells were detected and ready for analysis. Finally, 3D coordinates of the centroids of pyramidal cells were investigated for columnar patterns by using the cylindrical *K*-function.

*Annotation.* Data annotation is the method of labeling objects of interest that are detectable. The entire process for marking the pyramidal cells for our dataset was performed on 35 random cropped images from the aligned stack of each subject, is illustrated in Fig. S16. Here, the annotation was manually performed by an expert (NYL) using Photoshop's quick selection tool for each subject, but any image labeling program can be used, e.g., open-source tools such as VGG Image Annotator or ImageJ. Moreover, the Image Labeler program in MATLAB or employing the inbuilt function ginput to mark the border of a cell and then applying the imfill function to mask the cell may also be useful for labeling cells. Next, we used MATLAB to read the manually segmented (MS) images as binary images where pixels in pyramidal cell profiles equaled one, and all other pixel values equaled zero.

*Deep learning architecture.* Deep neural networks, in particular, convolutional neural networks (CNN), are commonly used for tasks of image classification[73]. For the research of this paper, we have chosen to continue working with the network described in the thesis 'Statistical analysis of pyramidal cells in brain tissue' where the code was published on GitHub[36]. The deep learning framework, UNetDense, is a modified version of the original UNet architecture and consists of several dense blocks, transition blocks and merging blocks adopted by the pre-trained DenseNet-121 to compute pixel-level predictions for neurons[74,75]. The Adam optimizer was used with a learning rate of $1 \times 10^5$, the loss function was Binary Cross Entropy plus Dice Loss, and the code was run on Google Colaboratory[36,76]. Cell profiles from 35 annotated images from each model were each sliced into 200 image patches of $256 \times 256$ pixels without redundancy, giving a total of 7000 ($35 \times 200$) images. The 7000 images were augmented by adjusting brightness and contrast and were then divided into a training set of 5600 images and a validation set of 1400 images. The differences in tissue from different subjects make it be difficult to train one combined model to segment the data of all subjects, see Table 2. Hence, we compared a combined model with individual models trained for each subject. Only the model for Subject 1 was trained from development, after which transferred learning was applied to the other two models afterward.

*Pixelwise validation.* We classified our predicted pixel values into four categories: true-positive (TP), false-positive (FP), true-negative (TN), and false-negative (FN).

TP: The total number of pyramid-pixels correctly identified by UNetDense model.
FP: The total number of pyramid-pixels wrongly identified by UNetDense model.
TN: The total number of non-pyramid-pixels correctly identified by UNetDense model.
FN: The total number of non-pyramid-pixel wrongly identified by UNetDense model.

For validation, the model was used to segment neurons from images that had also been manually segmented. The difference was then measured between the MS image and the UDP image, see Figs. S17–S19. The performance of each UNetDense model for segmenting 2D images was assessed using metrics of sensitivity, precision, and F1-score.

$$Sensitivity = TP/(TP + FN) \tag{1}$$

$$Precision = TP/(TP + FP) \tag{2}$$

$$F1 - score = \frac{2 \cdot (Sensitivity \cdot Precision)}{Sensitivity + Precision} \tag{3}$$

*Objectwise validation.* For validating the predictions of 3D-reconstructed cells, a new validation set was produced of 30 stacked images ($2048 \times 3840 \times 30$), which took about one week to complete. The data obtained was just a small portion of stacked images from Subject 1 and had not been seen by the UNetDense model before. The 3D-reconstruction from a stack of binary images and performance of the objectwise validation was done using custom MATLAB scripts, where we used the built-functions bwconncomp and regionprops3 for the 3D-reconstruction. After being reconstructed as 3D objects, the manually marked and predicted pyramidal cells were compared.

For this comparison, observed structures that did not appear in more than three consecutive images were omitted, corresponding to a height below 3 $\mu m$. Then, for both the MS and UDP images, we calculated the centroid of all detected pyramidal cells. A case where a MS centroid fell within a cell profile of the UDP 3D-reconstruction was denoted TP, while a case was denoted as FN when an MS centroid did not fall within such a cell profile. The FP case was defined to be the situation where a centroid of the UDP 3D-reconstruction did not fall within a cell profile of the MS 3D-reconstruction since these are falsely detected cells in the UDP. Based on these definitions, sensitivity, precision, and F1-scores were calculated.

The validation set contained 491 reconstructed and labeled pyramidal cells, which contains 5556 cell profiles in total, and we observed that the first and last

three images of the dataset predominantly contained false-negative neurons, see Fig. S20. The cause for this error was that centroids for cells extending beyond the image borders were poorly estimated. Thus, these were removed.

*Defining layer III.* The aligned segmented images from the output of the deep learning model were examined with custom MATLAB scripts. In this research, pyramidal cells in layer III were of primary interest. Thus, we identified a ROI containing only layer III by plotting a density map of a projection of estimated neuron centroids. These estimates were made from a total of 200 images loaded from the beginning, middle, and end of our complete aligned stack. We used the built-in function regionprops3 to estimate neuron centroid values from such images. Segmented 3D objects that were smaller than eight voxels were considered as artifacts and were thus filtered out just like centroid points from the first and last three images.

The density map depicts where the majority of cells were located, with yellow representing regions of high density and blue representing areas of low density. Layer III of the neocortex has a lower density than layer II and layer IV, then the ROI was specified between the two dense yellow areas for our analysis Fig. 4D. After a user had clicked on the top left and the bottom right corner to define the ROI, red dashed lines appeared to show the cropping frame, see Fig. 4D. This semi-automatic approach was chosen due to its reproducibility and effectiveness.

*3D-reconstruction and quantitative measurement of pyramidal cells.* The use of custom MATLAB scripts completed the study of morphological features and visualized the 3D-reconstruction of pyramidal cells, as shown in Fig. 5. Quantitative analytical values for each pyramidal neuron were approximated based on the entire stack of images, such as volume, centroid, diameter, maximum Feret diameter, orientation, surface area, and sphericity, where most values were calculated using the built-in function regionprops3.

Each cell's shape was assessed by approximating the sphericity, which does not have any prior assumption of shape and is independent of cell size. The sphericity is a dimensionless ratio, and the formula is given in Eq. (4), where *V* and *A* are the volume and surface area of the segmented cell. If a cell has sphericity equal to 1, it resembles a perfect sphere[37].

$$\psi = \frac{\pi^{1/3}(6V)^{2/3}}{A} \tag{4}$$

It was necessary to classify the detected neurons into pyramidal or non-pyramidal neurons as the UDP detects all neuronal shapes. This is because the top and bottom parts of a pyramidal cell have smaller profiles and can appear to be part of minor neurons or glial cells, as seen in Fig. S4.

A Gaussian mixture model (GMM) clustering algorithm was used to differentiate pyramidal and non-pyramidal neurons based on 3D measurements. The data used for the GMM consisted of the estimated volume and sphericity in 3D. The choice of data for the GMM is based on the idea that spherical objects with a smaller volume resemble smaller neurons or glial cells and hence non-pyramidal cells. For the dataset used for the GMM, we decided to only use the observed cells below the average volume of the dataset to ensure that only small round cells were evaluated. The dataset was fitted to the GMM using the default settings in the inbuilt fitgmdist function in MATLAB. After the GMM separated the dataset into two, the cells in the cluster with the lowest measurements were considered non-pyramidal cells and were excluded for all three subjects, see Fig. S5. Sometimes two cells were very close to each other and such merged cells were detected as one cell. Non-cellular structures like big vessels may also look like one cell. Such large, undesirable items were detected by identifying elements that had a log-transformed maximum Feret diameter measurement in 2D and 3D were greater than three standard deviations from the mean. This is particularly effective for distinguishing artifacts of unusual length because the maximum Feret diameter is the most extensive distance between two points in the convex hull, see Fig. S6 for the filtered data.

Finally, the orientation vector of a pyramidal cell is defined to be the unit vector *u* in the direction of the maximum Feret diameter, Eq. (5). The maximum Feret diameter provides information about the most extended length of a cell in a particular direction, usually toward the apical dendrite. The orientation angle θ is then defined to be the angle between *u* and the unit vector $u_0 = (1, 0, 0)$ which points in the direction of the *x*-axis. This means that if a pyramidal cell has orientation angle 0, its orientation vector points toward the pial surface. The orientation vector for each pyramidal cell can be calculated by

$$u = \frac{d - c}{||d - c||}, \tag{5}$$

where *c* and *d* are the vectors of the coordinates that define the maximum Feret diameter of the cell.

The orientation angle can be calculated by

$$\theta = \cos^{-1}(u_0 \cdot u) \tag{6}$$

*Quantification of pyramidal cell sizes in 2D.* The 2D analysis was performed on the precise same cells as the 3D analysis after removing non-pyramidal cells and outliers. Each identified cell consists of consecutive cell profiles that become a 3D cell entity after being combined, see Fig. S4. The volume of a given cell in 2D can be

estimated by using Eq. (7), where $\bar{l}$ equals the mean segment length from the centroid to the cell boundary, and $n$ equals the number of segments.

$$Volume = \frac{4}{3}\pi\bar{l}_n^3 \qquad (7)$$

This method is based on the well-known nucleator probe in stereology, which is used in biological research to estimate mean cell volume for quantitative histology[77]. This volume obtained by the nucleator probe relies on the mathematical fact that the mean intersection length, $\bar{l}_n$, between a unique point and the cell border by isotropic test lines can be viewed as a radius.

Two procedures were applied to test the 2D-analysis. For the first procedure, the largest cell profile was detected within the stack of profiles for each cell, which is usually near the middle of a cell. The nucleator probe was then applied on the largest cell profile and five-segment lengths were randomly positioned with a spacing between each other of 72° (360°/5), see Fig. S21. For the second procedure, the nucleator probe was employed on every profile of a cell and the average segment length was used to calculate the cell volume. The diameters for both procedures were estimated by Eq. (8).

$$Diameter = \bar{l}_n \cdot 2 \qquad (8)$$

*Point pattern analysis.* The statistical analysis was conducted with R Core Team (2019). For each of Subjects 1_1, 1_2, 2, and 3, the 3D coordinates for locations of the centroids of pyramidal cells in layer III of BA46 form a 3D spatial point pattern that was analyzed using statistical methods[78]. We used the cylindrical $K$-function[66], as was previously done in Rafi et al.[67] and Christoffersen et al.[68], to detect columnar structures in each 3D point pattern. In order to use the cylindrical $K$-function, we assumed that each point pattern was homogeneous. We assessed this assumption by looking at histograms of the projections of data onto the $x$-, $y$-, and $z$-axis and kernel smoothed intensity functions of the projections onto the $xy$-, $xz$, and $yz$-plane. Based on that, the point patterns seemed reasonably homogeneous. We denote the cylindrical $K$-function as $K_u(r, t)$ which makes it clear that it depends on a direction $u$ (a unit vector in 3D space), a radius $r$, and a height $t$, and we estimated it by the non-parametric approximation defined in Møller et al.[66]. Let $\rho$ be the intensity (mean number of points per volume unit). Then $\rho K_u(r, t)$ is interpreted as the expected number of further points inside a cylinder centered at a 'typical point' (intuitively a randomly selected point of the point process) with direction $u$, base radius $r$, and height $2t$, as exemplified in Fig. S22. If there is a columnar structure in a point pattern, the estimate of $K_u(r, t)$ is expected to be particularly high for the direction of the columnar structure for a range of $r$ and $t$ values. In order to decide whether $K_u(r, t)$ was significantly high, we compared it to the situation of Complete Spatial Randomness (CSR), meaning that there is no structure in data (a so-called homogeneous Poisson process), using a test called the extreme rank length global envelope test with a corresponding 95% global envelope[79,80]. This global envelope consists of a lower and upper curve such that the empirical cylindrical $K$-function for data falls entirely between these bounding curves if and only if the global envelope test cannot be dismissed at level 5% (more specifically at ~5% because we obtained the envelope based on simulations). When the empirical curve for $K_u(r, t)$ falls above the envelope, it means that it is higher than expected under CSR, which in turn indicates that there are cylinder-shaped clusters in the direction $u$. If the curve falls below the envelope, it indicates repulsive behavior between the points.

In our analysis, we looked at the directions corresponding to the three main axes, and we expected to find a columnar structure in the direction of the $x$-axis. We considered two situations for the global envelopes: First, we allowed $r$ and $t$ to vary and estimated the cylindrical $K$-function on a 64 × 64 grid where $r \in [0, 25]$ and $t \in [0, 80]$. We used 4000 simulations under CSR for the envelopes in this situation. Second, we fixed $t = 80$, meaning that $K_u(r, t)$ only depends on $r \in [0, 25]$. We estimated the function for 64 $r$-values and used 2000 simulations under CSR for the envelopes (These numbers of simulations follow the recommendations in the references above).

**Tissue deformation.** One biopsy with a diameter of 1.5 *mm* was taken from the gray matter of three human autopsy brains. The tissue area was carefully measured before and after it was dehydrated, embedded, sectioned, and stained. The area, $A$, of the tissue was estimated as:

$$A = \sum P \times (a/p)$$

where $\sum P$ is the number of test points hitting tissue and $(a/p)$ the area associated with each test point[38,39]. The areal shrinkage was estimated as:

$$Areal\ shrinkage = [(area\ before) - (area\ after)]/(area\ before) \qquad (9)$$

**Statistics and reproducibility.** Biopsies were obtained and examined from three subjects ($n = 3$), as stated throughout the article. The average, standard deviation, and coefficient of variation were used for the morphological measurements of pyramidal cells in layer III of BA46 for each subject. Calculations of the pixelwise and objectwise performance of the UNetDense architecture are described in the "Methods" section. Figures, tables, and histograms of this data were done using custom code via MATLAB. Statistical analysis of the spatial point pattern for each subject was performed using R Core Team (2019) and figures were produced using

the package ggplot2[81]. We employed the extreme rank length global envelope test with level 95% to determine whether $K_u(r, t)$ was significantly different from complete spatial randomness using 4000 simulations when constructing the envelopes. Statistical significance was defined as a *p*-value <0.05. All data sets presented in this work are available for download in our GitHub repository, https://doi.org/10.5281/zenodo.4287469[72,82].

**Reporting summary**. Further information on research design is available in the Nature Research Reporting Summary linked to this article.

## Data availability
The source data used to produce graphs and figures, and tables of quantitative measurements that support the current study's key findings are available from the corresponding author on reasonable request or in the GitHub repository, https://doi.org/10.5281/zenodo.4287469[72,82].

## Code availability
Source code of custom MATLAB and R scripts that support the findings of this study with image examples are available in the GitHub repository, https://doi.org/10.5281/zenodo.4287469[72,82].

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

## Acknowledgements

The brains analyzed in this paper are part of a brain series selected by Karl-Anton Dorph-Petersen. We want to thank Ute Hahn for providing us with the idea to sample the biopsies, Markus Kiderlen for help in estimating the orientation of pyramidal cells, and Hans Jacob Teglbjærg Stephensen for updating the deep learning code. We would also like to thank Jingyuan Yang and Mengyue Lou (FS lab) for their help in image

acquisition. We are grateful to Junfeng Hao from Laboratory Animal Facility Center, Institute of Biophysics, Chinese Academy of Sciences, for her use in light microscope management. We also want to thank Andreas Dyreborg Christoffersen for providing us with an R-function for estimating the cylindrical $K$-function. This work has been supported by the Centre for Stochastic Geometry and Advanced Bioimaging, funded by the Villum Foundation, Sino-Danish, and The Danish Council for Independent Research | Natural Sciences, grant DFF-7014-00074 'Statistics for point processes in space and beyond'. We also acknowledge the P30 GM103328 grant from the National Institutes of Health. We also thank grants from the National Natural Science Foundation of China (31925026) and the Chinese Academy of Sciences (QYZDB-SSW-SMC004).

## Author contributions

Conception and design: N.Y.L., F.S., and J.R.N.; sampling strategy and preparation: N.Y.L. and J.R.N.; optimizing the sampling procedure for the AutoCUTS-LM: N.Y.L., X.L., X.T., G.J., and F.S.; acquisition of data: N.Y.L. and X.T.; software/scripts developed for analyzing data: N.Y.L., J.L., J.S., and N.V.; analyzed the data: N.Y.L., J.S., N.V., and J.M.; supervised the project: N.Y.L., G.R., F.S., and J.R.N.; N.Y.L. took the lead in writing the manuscript, and N.V. contributed to the comprehensive revision of the article. All co-authors provided critical reviews and have accepted the final edition to be published.

## Competing interests

The authors declare no competing interests.
