## [Peer Review File · Communications Biology]

Reviewers' comments:

Reviewer #1 (Remarks to the Author):

In this paper, Larsen et al., described a new workflow for analyzing cellular architectures in the brain. They implemented advanced sampling procedures of biopsies from human cortices, automated collection of resin embedded brain sections, and deep learning-based segmentation of neurons. The application of ATUM method to LM samples is an innovative and feasible technique for many researchers. Detection of pyramidal cells from 3D reconstructed serial sections using a deep learning-based algorithm has the potential to become a useful tool for high throughput analyses. Overall, the techniques developed are well designed and address challenging issues in this field, however, the advantages and accuracy compared to current methods should be examined more carefully.

Major points:

1. To claim the advantages of their methods, the efficiency and accuracy of the pipeline should be compared with previous results. For example, the result shown in Table. 5 and the densities of pyramidal and non-pyramidal cells in layer III of BA46 should be compared with previous literatures and discussed carefully.
2. One possible analysis that might help emphasizing the advantage of the 3D analysis would be to compare 2D and 3D analysis using the exact same images.
3. The workflow for classifying pyramidal or non-pyramidal neurons is unclear. In figure 3, detection of pyramidal cells seems to be performed by UDP, however, in Fig. 5, cells were further classified into pyramidal or non-pyramidal neurons using the K-means algorithm. Since this manuscript is reporting a new pipeline, the workflow and description of each experiment need to be described more precisely in the main text.
4. Along the same line, the percentages of the cells omitted by K-means analysis because their large angle should be shown, and the relevance of the method should be discussed based on it. If the percentage is too high to be reasonably explained, their workflow needs to be improved.
5. The interpretation of K-means classification is difficult because the manuscript lacks detailed results of the classification. It is important to show the quantitative results of classification. Of the number of cells that were classified as non-pyramidal neurons, the authors should indicate the number that did not meet the requirements of radius and distance from the farthest voxel to the centroid, separately.
6. Since the 3D analyses were performed for stacks of 30 images, those stacks that are less than 30 micrometers thick should contain many cells that have only part of the cell body. Including those cells in the analysis would strongly affect the results. The authors should clarify if those cells were included in the analyses or not. If those cells are included in the analyses, such cells should be analyzed separately, and the results should be compared with those of the cells that have the entire cell body in the stack.

Minor point:

The resolution of figures should be higher for the publication. It was impossible to evaluate the EM images shown in Fig S4.

Reviewer #2 (Remarks to the Author):

This manuscript describes a full workflow for 3D reconstruction of pyramidal neuron soma from “punch biopsy” samples of the cerebral cortex of 3 human brains, including the automated serial sectioning, and automatic segmentation using a CNN, prepared by the authors. A brief morphological analysis of the shape of segmented neurons is also presented.

The text is well written and the data is well presented and very complete. The authors address many of the critical steps of the protocol, and optimizations they have done. The “tricks” shown here should be useful for those labs still relying heavily in histological techniques, but wanting to step into 3D analysis of cells and tissue organization.

The workflow involves several different steps, well described for sample preparation and tissue processing, sectioning, collection, image acquisition and analysis, and the authors have tried to automate most of the processes to minimize bias in sampling and analysis, with some exceptions (more in the comments section below). In GitHub the authors made the code developed publicly available and should be possible for others to reproduce the workflow either using the same software tools, or other (preferably) open-source tools. Because the authors are active on gitHub, it is likely that support will be available for those readers venturing to reproduce this protocol.

Pyramidal neurons are much more complex in shape and organization than what can be detected by this method, which only segments soma (the authors do acknowledge that the pyramidal cells can contain thousands or connections to other neurons), but, unless I missed it, no longer discuss that, in light of their findings, and the validity of making assessments on the analysis of only the soma of these really complex cells.

In each brain, 2 punched biopsies were taken and the authors do mention that only one was used for analysis, while the 2nd was stored. It is a pity that both were not analyzed, as this could provide an interesting control for the repeatability and accuracy of the workflow (assuming two samples of the same brain should produce consistent results).

I also sensed that this study lacked a comparison to other published protocols/methods so users can evaluate whether they should invest in implementing this technology, and all the resources it requires. For example, it is mentioned that tissue clearing techniques also have disadvantages, but the workflow here described is a lot more complex than tissue-clearing a 1-5mmx1.5mm block of brain tissue and imaging it at high resolution in confocal or two-photon, and potentially be able to more accurately measure the shape and organization of pyramidal cells: see for example a comparative study of mice and monkey in <https://doi.org/10.1093/cercor/bhw062>, or in mice in <https://doi.org/10.1038/s41598-020-64665-2>

Throughout the analysis the authors selective filter cells based on some parameters (eg, low sphericity,

“bad” orientation, “too” big) without explaining fully why. “Abnormally shaped” cells might actually be the interesting biology for other users trying to replicate this method. Perhaps this should be further discussed.

But I also see the value of it as a fully documented workflow, with important tips on how to improve histological analysis for human samples. In that sense, this paper reads more like a techniques, than a biology, communication.

Further comments:

Ln37 “Cylindrical” typo?

Ln67 Can the author’s explain better what they mean by “2D-histology dilemma for archival tissues”? Presumably they are referring to the fact that just from 2D sections, it is not easy to interpret 3D tissue organization and make accurate inferences about neuropathologies?

Ln158 repeated “)”)”

Ln211 Where the authors state “Hence, layer III should be between the two dense yellow areas in Fig. 4D” I think there needs a better explanation of how layer III was delimited. Fig4 legend is also not fully clarifying (what are the lines 1-7, why was limit between layer II and III delimited manually then?”

Ln214 “was used to categorise, pyramidal and non pyramidal neurons, see 3D-reconstruction of cells” incomplete sentence?

Discussion section:

Ln330 Here the authors say “The average sphericity of 0.35 found in this paper indicates that pyramidal cells appear to be far from spherical, “. Well, that was expected, no (they are pyramidal, afterall)? I think the authors need to be more explicit on how they evaluate this parameter, and how informative it can be, given that only soma are being measured- I comment more on this in the M&Ms section below. Furthermore, not sure I missed that info, but it was not said whether these 3 brains are of people without know pathologies? Can a comparison be made between them? What does the biology in the numbers presented tell us?

Ln347-353 Although it is exciting to see that “the results support the theory of a columnar patterns perpendicular to the pial surface”, the authors also recognize that larger numbers of samples are necessary. I feel a statistician should verify the validity of this sample and statistics and assert the validity of such a conclusion. It is also possible that this method does not yield reliable enough data that would warrant other researchers to adopt it.

Ln367... can the authors prescribe how this matching of the resin hardness to the tissue can be achieved?

Ln409 “The sampling area was divided into four quarters. “ why? what is the rationale for this procedure? Wouldn’t a skeletonization, representing the line of gyrus be easier? How accurate is the punching? Do authors have evidence that this is necessary and how would it impact the data if not done? It was not clear to me why this convoluted routine was necessary and what is gained?

Ln483 "...which were a few meters long..." is meters the word really intended here?

Ln 515 "(lens 20x, NA 0.8) with a resolution down to 272 nm." The claimed resolution seems hardly possible with that objective and without a high NA condenser. Authors probably meant pixel (or sampling) size.

Ln523 Can the authors comment more on their experience using this particular slide scanner and extracting data from the software, possibly as compared to other scanners/solutions available in the market. Also, on how easy it is to export files and their sampling size. Were these simple multi-megapixel TIFFs RGB files out of the Aperio software?

Ln525 Where is the script presented?

Ln527 "converted into a* grayscale" perhaps converted to grayscale?

Ln 529-530 "...to detect which section* were in focus..." sections? And it was not clear to me what is meant by "were in focus". Aren't all images supposed to be in focus? And if not, which were used then?

Ln563 "for *of cells" is a word missing here? Perhaps "all"?

Ln569 can the authors explain the choice of Photoshop in this step, and why not Matlab? Perhaps also recommend other tools for authors trying to reproduce the workflow without access to Photoshop (an open source tool like ImageJ comes to mind).

Ln588 Suggestion: given that the DL code was run on Google Colaboratory, would the authors consider contributing this to the community "ZeroCostDL4Mic"
<https://github.com/HenriquesLab/ZeroCostDL4Mic> to help make it more accessible (and thus increasing the visibility).

Ln 649-653 In this paragraph I think the author should explain better why this couldn't be automated as it was done for example in Fig 1E-F, to avoid the user bias in delimiting layer II-III

Ln 658 how is diameter in this context defined/measured?

Ln 666 can the authors comment on the advantage of using this vector between centroid and furthest voxel, instead of simple the cell's major axis length (3D Feret diameter)?

Ln 669 can the authors comment on choice of the "sphericity" instead of, for example, measuring "roundness" which seems more appropriate to characterize the roughed edges of the cell body of pyramidal neurons?

Ln675 despite having measured sphericity, the authors then proceed to classify neurons as pyramidal

based on a k-means analysis, based on average radius. What is then the use of the sphericity diameter? In this line, given that we're talking about 3D shapes, perhaps circular is not the best word, "more spherical" would be better.

Ln689 the choice of these two values, 40um and 30um for max length considered valid for a TP detection, should be better explained, and indications given for others trying to replicate the method. Different contrast/staining results and segmentation may produce reconstructed soma that are bigger or smaller. It wasn't completely clear to me that k-means clustering itself can be used to cluster these "outlier" detections as the authors mention. This would be a more unbiased method, and one that can guarantee reproducibility (instead of a viewer manually defining a min and max size).

Ln708 So far I had assumed this method detected only some and at most the basal part of the dendrites (basal and apical). Here the authors assume that for cells oriented parallel to the pial surface, that the orientation parameter was not correctly measure, and I presume those cells are eliminated from the analysis. The authors mention that those were removed, however looking at the graphs for orientation, there are several cells with an orientation close to 100, in all 3 samples.

Ln750 Where is the date on tissue deformation and how was it used? I presume the authors refer to a different sample taken from a different part of the brain.

Ln760 great that the authors are sharing the data, however I would suggest uploading to Zenodo or IDR, instead of an ephemeral google drive link.

Fig 1 in D) the legend boxes are unreadable, but instead of a yellow (as mentioned throughout the text), there is a green. In legend text where we read "were marked with a green line" perhaps replace with "green dashed line".

Fig3 where the authors say "where each second section was in focus..." I'm not sure "focus" is the best choice of word here. I think they meant to say "section of interest". Same happens below when the authors mention "areas in focus". In B)iii perhaps the numbers should not have blue boxes around, it's a bit confusing.

Fig4 I may have missed it, but why were those 6 lines drawn and how? What is the criterion to crop layer III to "line 6"? Also, why didn't the authors use some unbiased quantitative parameter (instead of having a user manually user defined limit), similar to the method used in Fig1E to sample the gyrus?

Table5 I understand the authors decided to log the values, given that the distributions were skewed, but why was this done? It gives the impression in the graphs in Fig5 that there is less variability than that that was actually found. Is the skewness caused by the "filtration" of unwanted cells (undesired size, shape and orientation)?

Also, the authors present a global mean that is the average of 3 averages, and present a small SD. I

believe these values should be checked by a statistician, as I suspect this is not an accurate representation of the data. Perhaps this calls for nested statistics.

In FigS4, just for accuracy, there is an image showing the use of antibodies, however in the Nature's reporting summary, use of antibodies were not considered for this study.

FigS5 "Diamter" is misspelled in the 2nd row of graphs. The use of log scales in X is unexpected. Perhaps a statistician should confirm that this is an acceptable representation of the data. It sure makes it look a lot less variable than it really is (and as can be seen in table5 with the large SDs reported).

Reviewer #3 (Remarks to the Author):

In this manuscript, Larsen and colleagues describe a pipeline to collect and image serial semi thin sections from fixed human cortex. They next use a neural network-based method to reconstruct in 3D the cortical samples.

Overall the text is too complex, full of technical jargon and I wonder who will want to use this method, at least routinely. Importantly, at the end of the paper, what have we learnt of the human cortex cytoarchitecture that was not already known?

First, the method only allows reconstructing a very small sample (only 700 μ m thick). The authors say that the biopsies contain the 6 layers of the cortex but they apparently need to remove the top and bottom layers of the cortex. Therefore, at this stage they seem unable to reconstruct a cortical column. Second, the identification of the cells/neurons, the quantification of their size and polarity only relies on simple toluidine blue staining (which is not precise) and it would be more useful if they could show that is compatible with immunolabelling (the GFAP immuno, Fig S7, is quite poor). Moreover, sections have many wrinkles (Fig. S3) and it is likely that their cannot be properly aligned (this should be demonstrated).

It also seems that not even all sections in the stack are imaged which again limit the interest of the pipeline.

Much better methods such as expansion microscopy are now available and give access to spines and synaptic contacts.

The authors mention clearing methods and claim that they do not work well with immunostaining of thick samples. However, there are many published studies, from the Ueda and Ertürk labs among others, which showed that large (centimeters) tissue pieces and even intact human organs can be immunostained and imaged in 3D.

Figure 4: a comparison of the raw images and binary ones should be shown

Figure 5: raw images, without pseudo colors should be shown

Figure 6: what does it mean? Do the authors think it is self explanatory?

Response to the referee reports on

Cellular 3D-reconstruction and analysis in Human Cerebral Cortex using Automatic Serial Sections

by

Larsen NY *et al.*

We would like to sincerely thank the referees for the suggestions for improvements of mainly the presentation of our results. We have carefully considered the comments by the referees and followed the suggestions by the referees except in very few cases, which are explained below. We believe that this has led to important improvements of the original paper, having in mind that the purpose of our paper is to make our method and data available for neuroscientists.

The comments by the referees are in bold and italics while our response is in normal font. The notations made in Latex are included in the answer since they were copied directly from the Latex file. However, page and line numbers are supplied in our response to the revised manuscript, so the identical paragraphs may be viewed without any code notations.

Referee 1

In this paper, Larsen et al., described a new workflow for analyzing cellular architectures in the brain. They implemented advanced sampling procedures of biopsies from human cortices, automated collection of resin embedded brain sections, and deep learning-based segmentation of neurons. The application of ATUM method to LM samples is an innovative and feasible technique for many researchers. Detection of pyramidal cells from 3D reconstructed serial sections using a deep learning-based algorithm has the potential to become a useful tool for high throughput analyses. Overall, the techniques developed are well designed and address challenging issues in this field, however, the advantages and accuracy compared to current methods should be examined more carefully.

1. ***To claim the advantages of their methods, the efficiency and accuracy of the pipeline should be compared with previous results. For example, the result shown in Table. 5 and the densities of pyramidal and non-pyramidal cells in layer III of BA46 should be compared with previous literatures and discussed carefully.***

A detailed response related to the density has been added to the discussion section.

Density (Ln462-488)

The observed pyramidal neuron density in layer $\text{Romannum}\{3\}$ of BA46 after the k-means clustering algorithm is $28160 \text{ } \mu\text{m}^{-3}$. A stereology study by Francine M. Benes (1986) found a neuronal density of $36800 \text{ } \mu\text{m}^{-3}$ in layer $\text{Romannum}\{3\}$ of postmortem brains from nine healthy controls in the prefrontal cortex \cite{Benes1986}. $\backslash\backslash$

Cullen et al. (2006) estimated a neuronal density of $37000 \text{ } \mu\text{m}^{-3}$ in the prefrontal cortex of 10 adults with no history of mental illnesses, similar to Benes et al. However, Cullen et al. also measured the density of pyramidal cells in layer $\text{Romannum}\{3\}$ to be $25650 \text{ } \mu\text{m}^{-3}$, a difference of 30% compared to the total neuronal density. $\backslash\backslash$

Rajkowska et al. (1995) characterized and mapped the cytoarchitecture of BA46 in 17 healthy people. They were able to estimate a neuronal density of $51510 \text{ } \mu\text{m}^{-3}$ by blending cortical layers $\text{Romannum}\{1\}$ - $\text{Romannum}\{3\}$ \cite{Rajkowska1995}. Because the density in layer $\text{Romannum}\{2\}$ varies from 48000 - $78000 \text{ } \mu\text{m}^{-3}$ and covers a significantly smaller area than layer $\text{Romannum}\{3\}$, it is reasonable to assume that the density in layer $\text{Romannum}\{3\}$ would be about 36000 - $44000 \text{ } \mu\text{m}^{-3}$ \cite{Benes1986,Cullen2006}. If the difference between neuronal and pyramidal density is roughly 30%, then the projected density should be approximately 25200 - $30800 \text{ } \mu\text{m}^{-3}$, which is consistent with our findings. $\backslash\backslash$

One thing all of the three studies listed above have in common is that they solely measured prefrontal cortex neuronal density \cite{Benes1986,Rajkowska1995,Cullen2006}.

Number densities indicate changes in the number of cells as well as tissue volume under consideration, because they are ratios.

It is commonly assumed that as the number of detected objects rises, so does the density.

However, the density increases as well if the number of identified objects remains constant, but the examined volume shrinks. As noted in other studies, making any definite claim regarding changes in 3D structures based purely on density estimates may lead to questionable findings \cite{Brendgaard1986, Swaab1987, Oorschot1994}. $\backslash\backslash$

Because this assumption is so widespread, the phrase "reference trap" refers to circumstances in which wrong conclusions are drawn only on the basis of density. As a result, comparing densities should be handled very carefully.

2. One possible analysis that might help emphasizing the advantage of the 3D analysis would be to compare 2D and 3D analysis using the exact same images.

A comparison between 2D vs 3D analysis have now been performed and new sections have been added in the revised paper. We had to update the content in the method, result, and discussion sections in the revised version because we did not compare 2D to 3D in the initial version.

Method (Ln871-894)

\subsubsection*{Quantification of pyramidal cells sizes in 2D}

The 2D analysis was performed on the precise same cells as the 3D analysis after removing non-pyramidal cells and outliers.

Each identified cell consists of consecutive cell profiles that become a 3D cell entity after being combined, see **\textbf{Fig. \ref{fig:3D_reconstruct}}**. The volume of a given cell in 2D can be estimated by using **\textbf{Eq.\ref{eq:volEst}}**, where \overline{l} equals the mean segment length from the centroid to the cell boundary, and n equals the number of segments.

$$\begin{equation} \text{Volume} = \frac{4}{3}\pi\overline{l}^3 \\ \text{\label{eq:volEst}} \end{equation}$$

This method is based on the well-known nucleator probe in stereology, which is used in biological research to estimate mean cell volume for quantitative histology **\cite{Gundersen1988}**. This volume obtained by the nucleator probe relies on the mathematical fact that the mean intersection length, \bar{l}_n , between a unique point and the cell border by isotropic test lines can be viewed as a radius.

Two procedures were applied to test the 2D-analysis. For the first procedure, the largest cell profile was detected within the stack of profiles for each cell, which is usually near the middle of a cell.

The nucleator probe was then applied on the largest cell profile and five segment lengths were randomly positioned with a spacing between each other of 72° ($360^\circ/5$), see **\textbf{Fig.\ref{fig:nucleator_crop}}**.

For the second procedure, the nucleator probe was employed on every profile of a cell and the average segment length was used to calculate the cell volume.

The diameters for both procedures were estimated by **\textbf{Eq.\ref{eq:Diameter}}**.

$$\begin{equation} \text{Diameter} = \overline{l}_n \cdot 2 \\ \text{\label{eq:Diameter}} \end{equation}$$

Result (Ln258-275)

2D vs 3D comparison of pyramidal cells sizes

The exact same neurons were used to compare the 2D and 3D analyses for each subject. The volumes of pyramidal cells from 2D images were approximated by constructing a spherical object based on the estimated radii measured by the nucleator probe.

The volumes of cells from the three subjects were calculated from 2D images using the estimated segment length from the largest cell profile (V_{L}) and all cell profiles (V_{All}).

The average neuronal volume in 3D (V_{3D}), V_{L} and V_{All} are 795, 730 and 183 μm^3 , respectively.

An approximation to the mean diameter of a cell D_{3D} was estimated from the mean volume V_{3D} under the assumption that it is the volume of a perfect sphere.

D_{3D} was then assessed and compared to the diameter measured from the largest cell profile (D_{L}) and all cell profiles (D_{All}). Because the nucleator probe is derived from the mathematical fact that length of isotropic test lines between a unique point and the cell border, it provides the most accurate one-to-one comparison. The average diameter for D_{3D} , D_{L} and D_{All} are 11.48, 11.17 and 7.03 μm , respectively.

The values for the 2D vs 3D comparison of volume and diameter can be seen in

Tab. ref{tab:vol}-ref{tab:dia}.

Discussion (Ln423-461)

The volumes of pyramidal cells calculated from 3D reconstruction images were compared to the results of the nucleator probe measurements for the same pyramidal cells, which estimate the mean cell volume from 2D images. It is crucial to keep in mind that the volume estimates are based on two separate methodologies, and the volume may be overestimated or underestimated with either approach.

The 3D approach measures every cell profile, and if a cell profile on the edge of a pyramidal cell is not recognized, this approach may underestimate the volume (top or bottom cell profile). On the other hand, the nucleator relies on measurements from a unique position (e.g. nucleolus), assuming that the section or cell is isotropic. If this criterion is not fulfilled, the estimate may be biased.

The average estimated volumes were 795, 730, 183 μm^3 , which gives a difference of 8.7% and 77%. It appears that using the nucleator probe on the largest cell profile to estimate volume yields a comparable overall result, but employing the nucleator probe on all cell profiles causes a huge difference. The estimated average volumes varies greatly depending on the approach used.

The estimated diameter changed less than the volume estimates, with results of 11.48, 11.17, and 7.03 μm , representing a 2.9% and 39% difference, respectively.

Hence, if we look at estimated radii when employing the largest cell profile compared to the 3D estimation, there is no substantial disagreement between those two approaches for calculating diameter because the difference is nearly equivalent to the 272 nm picture resolution. In contrast, estimating the average radius from all cell profiles makes a significant difference. As the neurons are not spherical in reality, comparing neuronal size based on volume rather than diameter is a more appropriate parameter to quantify changes in neuronal size in future references. Nevertheless, if only a 2D technique is available, adopting a spherical approximation by assuming general isotropy of these human pyramidal cells to estimate neuronal volume is not

entirely inaccurate. It is vital to notice that a slight change in the radius can substantially change the volume value considering the radius is raised to the power of three while computing the volume.\\

Even though the nucleator is a 2D procedure, it is nevertheless based on a 3D sampling methodology since the biggest cell profile detected during the sample is required before the analysis can be completed. The nucleator is therefore a helpful tool to use when it is challenging to distinguish smaller cell profiles at the edges and only larger cell profiles are required. \\

Another important consideration is that the nucleator will be less sensitive to tissue shrinkage in the direction of the z-axis. The nucleator is advantageous for estimating volumes compared to other embedding materials that are sensitive to shrinkage, such as frozen sections or vibratome sections, which rarely deform in the x- and y-axes but shrink in the z-axis.

- 3. *The workflow for classifying pyramidal or non-pyramidal neurons is unclear. In figure 3, detection of pyramidal cells seems to be performed by UDP, however, in Fig. 5, cells were further classified into pyramidal or non-pyramidal neurons using the K-means algorithm. Since this manuscript is reporting a new pipeline, the workflow and description of each experiment need to be described more precisely in the main text.***

The detection/segmentation of pyramidal cells is trained from sections that are 2D images. Try to imagine a pyramidal cell is shaped like a cone. If a cone/pyramidal cell is cut near the edge, it will be projected as a circular shape and not triangular. The segmentation is therefore also designed to detect smaller profiles of neurons and/or glial cells, since it needs to detect these profiles in order to recreate the cone/pyramidal cell.

That is why an extra filtration is necessary to remove other shapes because the UDP did detect all cells. Figure S4 is added in the supplementary to illustrate why further classification is needed based on their 3D shape and not 2D images.

In the revised paper this clarification is made clear in the following sections:

Result section (Ln223-237)

Further classifications into pyramidal or non-pyramidal neurons were needed as the UDP detects all neuronal shapes from 2D images. This is necessary because pyramidal cells' top and bottom image profiles can be mistaken for smaller neurons/glial cells, as seen in \textbf{Fig. \ref{fig:3D_reconstruct}}.

The K-means clustering algorithm was used to classify the 3D-reconstructed pyramidal and non-pyramidal cells based on estimated sphericity and volume, see \textbf{Fig. \ref{fig:neurons3D}B}. Big objects/cells were classified as outliers if the maximum Feret Diameter in 2D or 3D measurements was three standard deviations from the mean. As a result, a total of 1, 19 and 28 datapoints for each subject were deemed outliers and hence excluded from the pyramidal cell data. The percentage of objects/neurons excluded from analysis using K-means analysis and outlier detection accounted for 23%, 25%, 37%, and 23% of the total number of detected objects for each dataset, as shown in \textbf{Tab. \ref{tab>window}}.\\

The mean density of pyramidal cells in layer \Romannum{3} of BA46 after filtering was $28155 \mu\text{m}^{-3}$, and the K-means segmentation and data containing outliers are shown in \textbf{Fig. \ref{fig:kmeans_plot1}-\ref{fig:kmeans_plot2}}.

Method section(Ln835-859)

It was necessary to classify the detected neurons into pyramidal or non-pyramidal neurons as the UDP detects all neuronal shapes. This is because the top and bottom parts of a pyramidal cell have smaller profiles and can appear to be part of minor neurons or glial cells, as seen in \textbf{Fig.\ref{fig:3D_reconstruct}}.\\

The k-means algorithm was used to differentiate pyramidal and non-pyramidal neurons based on 3D measurements.\\

The data used for k-means consisted of the estimated volume and sphericity in 3D. The choice of data for k-means is based on the idea that spherical objects with a smaller volume resemble smaller neurons or glial cells and hence non-pyramidal cells. For the dataset used for k-means, we decided to only use the observed cells below the average volume of the dataset to ensure that only small round cells were evaluated.\\

The data set was partitioned into two clusters using k-means, and the data were fitted to a Gaussian mixture model using the default settings in the inbuilt fitgmdist in MATLAB. After k-means divided the dataset into two, the cells in the cluster with the lowest measurements were considered non-pyramidal cells and were excluded for all three subjects, see \textbf{Fig.\ref{fig:kmeans_plot1}}.

Sometimes two cells were very close to each other and such "merged" cells were detected as one cell. Non-cellular structures like big vessels may also look like one cell. Such large, undesirable items were detected by identifying elements that had a log-transformed maximum Feret diameter measurement in 2D and 3D were greater than three standard deviations from the mean.\\

This is particularly effective for distinguishing artifacts of unusual length because the maximum Feret diameter is the most extensive distance between two points in the convex hull, see \textbf{Fig.\ref{fig:kmeans_plot2}} for the filtered data.\\

- 4. Along the same line, the percentages of the cells omitted by K-means analysis because their large angle should be shown, and the relevance of the method should be discussed based on it. If the percentage is too high to be reasonably explained, their workflow needs to be improved.***

A thorough response to the percentages of cells omitted by K-means analysis has been added in Question #3, referee #1. The density of pyramidal cells matches with the literature after filtering of non-pyramidal cells and has been added in the discussion section. The number of excluded cells and densities for the three individuals are shown in Table 4.

Density (Ln462-488)

The observed pyramidal neuron density in layer $\text{Romannum}\{3\}$ of BA46 after the k-means clustering algorithm is $28160 \text{ } \mu\text{m}^{-3}$. A stereology study by Francine M. Benes (1986) found a neuronal density of $36800 \text{ } \mu\text{m}^{-3}$ in layer $\text{Romannum}\{3\}$ of postmortem brains from nine healthy controls in the prefrontal cortex \cite{Benes1986}. \\

Cullen et al. (2006) estimated a neuronal density of $37000 \text{ } \mu\text{m}^{-3}$ in the prefrontal cortex of 10 adults with no history of mental illnesses, similar to Benes et al. However, Cullen et al. also measured the density of pyramidal cells in layer $\text{Romannum}\{3\}$ to be $25650 \text{ } \mu\text{m}^{-3}$, a difference of 30% compared to the total neuronal density.\\

Rajkowska et al. (1995) characterized and mapped the cytoarchitecture of BA46 in 17 healthy people. They were able to estimate a neuronal density of 51510 mm^{-3} by blending cortical layers 1-3 (Rajkowska1995). Because the density in layer 2 varies from 48000-78000 mm^{-3} and covers a significantly smaller area than layer 3, it is reasonable to assume that the density in layer 3 would be about 36000-44000 mm^{-3} (Benes1986,Cullen2006). If the difference between neuronal and pyramidal density is roughly 30%, then the projected density should be approximately 25200-30800 mm^{-3} , which is consistent with our findings.

One thing all of the three studies listed above have in common is that they solely measured prefrontal cortex neuronal density (Benes1986,Rajkowska1995,Cullen2006).

Number densities indicate changes in the number of cells as well as tissue volume under consideration, because they are ratios.

It is commonly assumed that as the number of detected objects rises, so does the density. However, the density increases as well if the number of identified objects remains constant, but the examined volume shrinks. As noted in other studies, making any definite claim regarding changes in 3D structures based purely on density estimates may lead to questionable findings (Brendgaard1986, Swaab1987, Oorschot1994).

Because this assumption is so widespread, the phrase "reference trap" refers to circumstances in which wrong conclusions are drawn only on the basis of density. As a result, comparing densities should be handled very carefully.

5. ***The interpretation of K-means classification is difficult because the manuscript lacks detailed results of the classification. It is important to show the quantitative results of classification. Of the number of cells that were classified as non-pyramidal neurons, the authors should indicate the number that did not meet the requirements of radius and distance from the farthest voxel to the centroid, separately.***

As mentioned in Question #3, referee #1, the classification results are presented in Fig S6 and the number that did not meet the requirement is shown in table 4.

6. ***Since the 3D analyses were performed for stacks of 30 images, those stacks that are less than 30 micrometers thick should contain many cells that have only part of the cell body. Including those cells in the analysis would strongly affect the results. The authors should clarify if those cells were included in the analyses or not. If those cells are included in the analyses, such cells should be analyzed separately, and the results should be compared with those of the cells that have the entire cell body in the stack.***

We are sorry, but this is a misunderstanding. The 3D-analysis was carried out on the entire stack of aligned images, 800-1100 images, and not on only 30 images. The example with 30 images in figure S15 was to validate the output of the UNetDense network. We labelled 491 pyramidal cells manually, which included all their cell profiles (5556) through all 30 images. To see whether the same pyramidal cells were found, those manually segmented images were then compared with the output of the UNetDense network. The centroid calculation of the first three and last three images was omitted for any stack, as FN values were observed. This is because cells contain only portions of the observed pyramidal cells at the beginning and end of any tissue stack and not the whole cell.

In the revised paper this clarification is made clear.

Result section (Ln238-240)

The measurements and calculations for each subject's entire stack were examined just for the classified pyramidal cells, with the number of cells and the size of the ROI window for each subject shown in table \ref{tab:window}.

Method section (Ln826-829)

Quantitative analytical values for each pyramidal neuron were then approximated based on the entire stack of images, such as volume, centroid, diameter, maximum Feret diameter, orientation, surface area, and sphericity, where most values were calculated using the built in function `regionprops3`.

7. ***The resolution of figures should be higher for the publication. It was impossible to evaluate the EM images shown in Fig S4.***

The document was compressed from a 60MB file to less than 25MB because 30 MB is the maximum file size that was written to be submitted on the website. The original EM image is attached in the revised paper (without compression) and it has better resolution. In the revised manuscript, S4 has been changed to S3.

Referee 2

This manuscript describes a full workflow for 3D reconstruction of pyramidal neuron soma from “punch biopsy” samples of the cerebral cortex of 3 human brains, including the automated serial sectioning, and automatic segmentation using a CNN, prepared by the authors. A brief morphological analysis of the shape of segmented neurons is also presented.

The text is well written and the data is well presented and very complete. The authors address many of the critical steps of the protocol, and optimizations they have done. The “tricks” shown here should be useful for those labs still relying heavily in histological techniques, but wanting to step into 3D analysis of cells and tissue organization.

The workflow involves several different steps, well described for sample preparation and tissue processing, sectioning, collection, image acquisition and analysis, and the authors have tried to automate most of the processes to minimize bias in sampling and analysis, with some exceptions (more in the comments section below). In GitHub the authors made the code developed publicly available and should be possible for others to reproduce the workflow either using the same software tools, or other (preferably) open-source tools. Because the authors are active on gitHub, it is likely that support will be available for those readers venturing to reproduce this protocol.

- 1. Pyramidal neurons are much more complex in shape and organization than what can be detected by this method, which only segments soma (the authors do acknowledge that the pyramidal cells can contain thousands or connections to other neurons), but, unless I missed it, no longer discuss that, in light of their findings, and the validity of making assessments on the analysis of only the soma of these really complex cells.***

Maybe there is a misunderstanding here. It is correct that pyramidal cells can have thousands of synaptic connections to other neurons, however, we are not interested or able to detect synapses with this resolution and synapses have never been part of such cell columnarity analysis. Therefore, we are only interested in the detection of the placement and structure of somas of human pyramidal cells, which is doable with the use of this Epon embedding, 3-400 nm thick serial sections and a 20X (NA 0.8) objective mounted on a modern Leica light microscope.

This is obvious, when you look at images in figures 5 and 8, where the typical average “diameter” of these complex cells are 11-13 μm and the pixel resolution is 272 nm.

In the revised paper this is pointed out in Ln58-71 in the introduction:

Introduction - Updated text (Ln58-71)

The cellular organization in the human neocortex has been described as a local network of vertical columns known as 'minicolumns,' and such a column is regarded as the smallest unit capable of information processing in the cerebral cortex\cite{Mountcastle1957}. Minicolumns are radially oriented cell bodies that span through the laminar pattern perpendicular to the pial surface and can be seen using regular Nissl preparations or another cell body-revealing histological procedures.

The introduction of minicolumns was in response to studies of the patterning of apical dendrites of pyramidal cells with somata situated in layers \Romannum{2}, \Romannum{3}, and

\Romannum{5}\cite{Fleischhauer_1972,Peters1972}. Studies have attempted to characterize and analyze the morphology of minicolumns with a 2D computerized method designed to detect subtle differences among patient groups such as schizophrenia, autism, and Alzheimer\cite{Chance2008,Casanova2003,McKavanagh2015,Raghanti2010}. As a result, much of our understanding of cellular organization is focused on 2D histological images, which could potentially misrepresent biological structures and malpositioning of cells in 3D-space.

- 2. In each brain, 2 punched biopsies were taken and the authors do mention that only one was used for analysis, while the 2nd was stored. It is a pity that both were not analyzed, as this could provide and interesting control for the repeatability and accuracy of the workflow (assuming two samples of the same brain should produce consistent results). I also sensed that this study lacked a comparison to other published protocols/methods so users can evaluate whether they should invest in implementing this technology, and all the resources it requires.**

The 2nd biopsy was a backup in case of errors due to processing, cutting and staining and we do not have the other biopsies due to cutting failures in the start of the study as the study actually used the 2nd biopsy.

However, we have also embedded the remaining part of BA46 into bigger blocks of glycolmethacrylate for another study. Here, we have also estimated the 3D number density of pyramidal cells in layer 3, BA46 with the optical fractionator/Cavalieri estimator using the remaining block of tissue from the same 3 autopsy brains, Fig.1B (old document). Every second of 2.5 mm thick slabs through the whole of BA46 was sampled, embedded in glycolmethacrylate, cut into 40 μm thick sections and stained with Nissl. Using an Olympus BX51 microscope modified for stereology with a Prior motorized stage, Heidenhain z-encoder, Basler USB3 digital camera and connected to a PC with newCAST software, the sections were sampled with 1040 μm in both x- and y-direction with a 2D counting frame of 3244 μm^2 . The optical disector was applied following a z-axis analysis in the central 20 μm of the section. The numerical density of pyramidal cells was $\sim 30.000 \text{ neurons}_{\text{PYR}} / (10^3 * \text{mm}^3)$, which is similar to the ~ 28.000 we estimated in this current study.

Having access to older chemical fixed human autopsy brains, this paper describes a method for analyzing unique parameters of single neurons/cells and columnarity of neurons/cells. Methods relying on antibodies are difficult to use on older chemical fixed human autopsy brains but if the antigens of the tissue are not destroyed our method can also be used together with antibodies as shown in Fig. S3A. Similarly, if higher resolution is needed, our method can also be combined with electron microscopy. Our old chemical fixed human autopsy brains could also be used for the original AutoCUTS method, <https://doi.org/10.1016/j.jsb.2017.09.010>, as well as the Automated Tape-Collecting Ultramicrotome (ATUM) method, <https://doi.org/10.3389/fncir.2014.00068>, however, these methods can only investigate a very small field of view, which would not have made the analysis of cell columnarity possible.

3. ***For example, it is mentioned that tissue clearing techniques also have disadvantages, but the workflows here described is a lot more complex than tissue-clearing a 1-5mmx1.5mm block of brain tissue and imaging it at high resolution in confocal or two-photon, and potentially be able to more accurately measure the shape and organization of pyramidal cells: see for example a comparative study of mice and monkey in <https://doi.org/10.1093/cercor/bhw062>, or in mice in <https://doi.org/10.1038/s41598-020-64665-2>***

There is not currently a specific antibody against (human) pyramidal cells. It is also quite difficult to make neuronal antibodies (NeuN, MAP-2, MBP, Nestin, Neurofilament _2F11, Neurofilament_DA2/FNP7/RmdO20.11) work in autopsy brains placed in fixative for months/years ([10.1369/jhc.7A7187.2007](https://doi.org/10.1369/jhc.7A7187.2007)). Even though a neural antibody would work in older autopsy brains, it would still be very difficult to be certain that ALL pyramidal cells were labelled. This is much easier to guarantee with Nissl staining. This is also why clearing techniques would be impossible/difficult to use with these old autopsy brains and that is why we have chosen Epon embedding, thin serial sectioning and toluidine blue staining.

Discussion (Ln338-347)

The majority of the approaches discussed above share the use of immunolabeling to detect and localize antigens or proteins within a cell at a specific location.\\

The fundamental constraint of immunolabeling is that milder fixation conditions and a shorter detergent permeabilization incubation time are required to allow antibodies to penetrate a cell or tissue. This is particularly important for soluble proteins in the cytoplasm, which are frequently damaged or destroyed during incubation. Hence, it is challenging to create neuronal antibodies that work in postmortem brains that have been in the fixative for months or years\cite{Lyck2007}.\\

Researchers have a unique opportunity as well as a terrible difficulty with archived tissue samples, enabling researchers to study disease development in great detail.

4. ***Throughout the analysis the authors selective filter cells based on some parameters (eg, low sphericity, "bad" orientation, "too" big) without explaining fully why. "Abnormally shaped" cells might actually be the interesting biology for other users trying to replicate this method. Perhaps this should be further discussed.***

But I also see the value of it as a fully documented workflow, with important tips on how to improve histological analysis for human samples. In that sense, this paper reads more like a techniques, than a biology, communication.

Selective filters have been excluded in the revised paper and the filtration is only dependent on the k-means and estimating outliers that are three standard deviations from the mean of the measured maximum Feret diameter. A detailed response can be seen in Question#3, referee#1.

No "bad" orientation does not appear anymore after we used the Feret Diameter to estimate the orientation of cells. It is an excellent recommendation from reviewer 2 to use this parameter. The maximum Feret diameter shows the object's natural orientation and can also be used to detect some objects with abnormal lengths/sizes. Our paper is implementing newly developed methods for the use in neuroscience trying to emphasize and broaden the applicability of old archived autopsy-based material to study important neuronal quantities and their spatial distribution.

5. ***Ln67 Can the author's explain better what they mean by "2D-histology dilemma for archival tissues"? Presumably they are referring to the fact that just from 2D sections, it is not easy to interpret 3D tissue organization and make accurate inferences about neuropathologies?***

Yes, we agree 100% with this statement of the reviewer – thank you.

In the revised paper this clarification is made clear in the introduction:

Introduction (Ln72-74)

This paper aims to create a method that is accessible to the broader science community and analyze 3D tissue organization through the use of archival tissue to make detailed inferences about pathologies.

6. ***Ln158 repeated ")]"***

The sentence in the revised article.

Results (Ln164-167)

Sections were then stained with toluidine blue, and the tape with attached sections was cut into three consecutive strips with approximately $60(20 \times 3)$ serial sections and glued onto a standard microscope slide `\textbf{(Fig.\ref{fig:AutoCutTapePrep}B-E)}`.

7. ***Ln211 Where the authors state "Hence, layer III should be between the two dense yellow areas in Fig. 4D" I think there needs a better explanation of how layer III was delimited. Fig4 legend is also not fully clarifying (what are the lines 1-7, why was limit between layer II and III delimited manually then?"***

A code to delineate layer III has been modified in the revised paper. The ROI is determined by an individual clicking on the top left and the bottom left corner. The figure text has now been updated with a better explanation in 4E.

Results (Ln216-221)

Layer `\Romannum{3}` was located using a density plot of a 2D-projection of the centroids of every 3D-reconstructed neuron, with yellow representing areas of high density and blue representing areas of low density, see `\textbf{Fig. \ref{fig:layer3analysis}}`. Layer `\Romannum{1}` got a very low density of neurons, while `\Romannum{2}` and `\Romannum{4}` are denser than layer `\Romannum{3}` in BA46. As Layer `\Romannum{3}` has a smaller density than Layer `\Romannum{2}` and Layer `\Romannum{4}`, the ROI is specified between the two dense yellow areas for our analysis.

8. ***Ln214 "was used to categorise, pyramidal and non pyramidal neurons, see 3D-reconstruction of cells" incomplete sentence?***

Thank you, the sentence has been updated in the revised paper.

Result (Ln226-228)

The k-means clustering algorithm was used to classify the 3D-reconstructed pyramidal and non-pyramidal cells based on estimated sphericity and volume, see \textbf{Fig. \ref{fig:neurons3D}B}.

Discussion section:

- 9. Ln330 Here the authors say “The average sphericity of 0.35 found in this paper indicates that pyramidal cells appear to be far from spherical, “. Well, that was expected, no (they are pyramidal, afterall)? I think the authors need to be more explicit on how they evaluate this parameter, and how informative it can be, given that only soma are being measured- I comment more on this in the M&Ms section below.**

You are right that measurement of sphericity of pyramidal cells confirm that they are far from spherical as expected. However, it could be interesting, whether it would change between different patient groups, which further studies will try to accomplish.

- 10. Furthermore, not sure I missed that info, but it was not said whether these 3 brains are of people without know pathologies? Can a comparison be made between them? What does the biology in the numbers presented tell us?**

The three brains are from dead people without known brain disorders. They can be regarded as controls and can be compared. This is a presentation of new methodologies for use with archival tissue in the brain and we cannot make any firm biological conclusions from just three brains. We and others will implement this methodology on many more brains in a clinical study, so it is possible to analyse whether or not quantitative brain pathologies are present. A remark about this has been inserted in the revised paper, see Ln546-550.

Materials and methods (Ln546-550)

Three healthy human brains (two women, one male) aged 30, 53 and 58 (Subjects 1, 2 and 3) with no history of psychiatric or neurological diseases were selected from the brain collection at Core Centre for Molecular Morphology, Section for Stereology and Microscopy, Aarhus University Hospital, Aarhus, Denmark.

- 11. Ln347-353 Although it is exciting to see that “the results support the theory of a columnar patterns perpendicular to the pial surface”, the authors also recognize that larger numbers of samples are necessary. I feel a statistician should verify the validity of this sample and statistics and assert the validity of such a conclusion. It is also possible that this method does not yield reliable enough data that would warrant other researchers to adopt it.**

This is a presentation of new methodologies for use with archival tissue using only three human autopsy brains, so all biological findings are preliminary. Two of our co-authors (Jesper Møller and Ninna Vihrs) are statisticians and performed the columnar structure analysis using spatial statistical methods (cylindrical K-function). Jesper Møller is an international leading authority in spatial statistics and he invented the cylindrical K-function for a paper published in *Biometrika*,

which is the most prestigious journal in the field of statistics. In short the statistical analysis in our paper is valid since the study was performed on thousands of data points for each subject using well documented statistical methods.

In fact, previous studies published in international statistical journals using the cylindrical K-function was performed using much less points; we have now stressed this at Ln490-Ln493

Discussion (Ln496-499)

The spatial distribution of pyramidal neurons was analyzed with the use of the cylindrical K_S -function, which does not assume any columnarity a priori, and where we applied the cylindrical K_S -function on much larger point pattern datasets than so far analyzed in the literature \cite{Moller2016,Rafati2016,Christoffersen2019}.

12. Ln367... *can the authors prescribe how this matching of the resin hardness to the tissue can be achieved?*

It is best to do a resin ratio test to match the tissue density and select the most suitable resin hardness before collecting new samples with AutoCUTS-LM. Generally speaking, by increasing and decreasing the ratio of two different anhydride curing agents (Dodecenyl Succinic Anhydride and NADIC Methyl Anhydride), any hardness can be achieved to suit a particular tissue type.

To achieve the resin hardness that is most appropriate for flattening our tissue sections, we have recently used this resin ratio (21 ml SPI-PON812, 13 ml DDSA and 10 ml NMA).

The revised paper now emphasizes how matching of tissue density and resin hardness can be achieved in the discussion.

Discussion (Ln520-525)

Applying a resin ratio test to match the tissue density and select the most suitable one before collecting new samples with AutoCUTS-LM would have a beneficial effect on reducing the folds in the section. Reducing the hardness of the resin so it suits a particular tissue type can be achieved by increasing and decreasing the ratio of two different anhydride curing agents (Dodecenyl Succinic Anhydride and NADIC Methyl Anhydride).

13. Ln409 *“The sampling area was divided into four quarters. “ why? what is the rationale for this procedure? Wouldn’t a skeletonization, representing the line of gyrus be easier? How accurate is the punching? Do authors have evidence that this is necessary and how would it impact the data if not done? It was not clear to me why this convoluted routine was necessary and what is gained?*

This method's rationale is to avoid overlapping the biopsies and ensuring that the biopsies are properly positioned at the right location without observer bias. The sample can only be carried out in areas between the 1st and 3rd quarters or 2nd and 4th quarters to avoid any overlap. It might be easier to implement skeletonization as you state, but some gyri are thicker than others, and it will exclude larger sampling areas if only the midpoint can be chosen.

14. Ln483 “...which were a few meters long...” is meters the word really intended here?

If we say each section has a length of 1.5 mm (layer I-IV) and there is 1 mm space between sections then one section covers around 2.5 mm. For each brain we cut around 2400 sections, which is a length of $2400 \times 2.5 \text{ mm} = 6000 \text{ mm} = 6 \text{ m}$ in total.

15. Ln 515 “(lens 20x, NA 0.8) with a resolution down to 272 nm.” The claimed resolution seems hardly possible with that objective and without a high NA condenser. Authors probably meant pixel (or sampling) size.

Adjusted to 272 nm pixel resolution.

Method (Ln673-674)

Next, the microscope collected images at higher magnifications (lens 20x, NA 0.8) with a pixel-resolution down to 272 nm.

16. Ln523 Can the authors comment more on their experience using this particular slide scanner and extracting data from the software, possibly as compared to other scanners/solutions available in the market. Also, on how easy it is to export files and their sampling size. Were these simple multi-megapixel TIFFs RGB files out of the Aperio software?

Our experience with the Leica scanner is generally good. After the focus points are placed on a glass slide, it is fully automatic without any supervision. Hence, it runs overnight and time is therefore saved during data collection. It is also easy to export files with the software compared to another microscope/slide scanner we have used in other studies (Olympus VS120). The output from the TIF files for each glass slide was around 6-9 GB in total (can be exported into smaller segments for each glass slide). However, those uncompressed TIF files have to be manually exported, which is why we made the script to perform the steps in Fig. 3. Ideally, a scanner should capture pictures and export them as tif files automatically. In that case, everything will be fully automatic since the script would load the RGB TIFF files and separate them into individual sections.

17. Ln525 Where is the script presented?

The scripts are presented at: <https://github.com/Nick7900/AutoCUTS-LM-Analysis>

18. Ln527 “converted into a* grayscale” perhaps converted to grayscale?

Thanks for your attentive reading. It is corrected.

Method (Ln689-690)

First, the large image files were converted to grayscale, followed by a Gaussian blur.

19. Ln 529-530 “...to detect which section* were in focus...” sections? And it was not clear to me what is meant by “were in focus”. Aren’t all images supposed to be in focus? And if not, which were used then?

We did sample either every second or every third image with the scanner as it resulted in a total thickness of either 800 nm (400 nm*2) or 900 nm (300 nm*3) between sections to reduce processing time, as a thickness below 1 µm is sufficient to reconstruct cells. Hence, only the sections we sample are in focus. Please see Fig 3 with a thickness of 800 nm distance between each image. This is stated in the old version (Ln516-518) and Ln674-677 in the revised paper.

Methods (Ln674-677)

Systematic sampling of our sections was done by only chosen every second (subject 1) and every third section (subject 2 and 3). This choice was based on the chosen cutting thicknesses of 400 nm and 300 nm, which correspond to a sample interval of 800 and 900 nm, respectively.

20. Ln563 “for *of cells” is a word missing here? Perhaps “all”?

Again thanks for your attentive reading. It is corrected.

Updated Text (Ln724-726)

Then, we performed 3D-reconstruction and calculated morphological parameters for all cells from the segmented neurons in layer \Romannum{3}.

21. Ln569 can the authors explain the choice of Photoshop in this step, and why not Matlab? Perhaps also recommend other tools for authors trying to reproduce the workflow without access to Photoshop (an open source tool like ImageJ comes to mind).

Thank you for your comment; Photoshop was chosen since labeling pyramidal cells were simple and we have access to this software. The quick selection method is great to get the whole-cell profile with minimal effort. Because we have limited experience with ImageJ, we did not use it to label the cells in this work. However, the Image Labeler app in MATLAB or using the inbuilt function `ginput` to mark the boundary of a cell and then apply `imfill` function to mask the cell may be useful. Some researchers use VGG Image Annotator (VIA) to annotate cells, which is a free online software tool (<https://www.robots.ox.ac.uk/~vgg/software/via/>).

A mention of additional open-source tools or other approaches for labeling images have been inserted in the revised paper, see Ln729-734.

Method (Ln735-740)

Here, the annotation was manually performed by an expert (NYL) using Photoshop's quick selection tool for each subject, but any image labelling program can be used, e.g. open-source tools such as VGG Image Annotator or ImageJ. Moreover, the Image Labeler program in MATLAB or employing the inbuilt function `ginput` to mark the border of a cell and then applying the `imfill` function to mask the cell may also be useful for labeling cells.

- 22. Ln588 Suggestion: given that the DL code was run on Google Colaboratory, would the authors consider contributing this to the community “ZeroCostDL4Mic”**

<https://github.com/HenriquesLab/ZeroCostDL4Mic> to help make it more accessible (and thus increasing the visibility).

The UNetDense Model is already available in GITHUB, but it needs a description update as it was performed on old test images from 2018: <https://github.com/JingLin0/Pyramidal-Cells-Segmentation>. We will like to contribute the code to the community “ZeroCostDL4Mic” following acceptance of the paper.

- 23. Ln 649-653 In this paragraph I think the author should explain better why this couldn't be automated as it was done for example in Fig 1E-F, to avoid the user bias in delimiting layer II-III**

One of the co-authors, Grazyna Rajkowska, is world famous for her ability to recognize and delineate BA46 in human autopsy brains. This is very challenging! Making an automatic procedure for this is difficult, so we prefer it to be semi-automatic, where a user choose the ROI. The extra explanation is added in the revised paper.

Original text

Layer \Romannum{3} of the neocortex were then identified by a density map of a projection of the estimated centroids in the following way (see also \textbf{Fig.\ref{fig:layer3analysis}D}). The starting point from which to crop the images was identified by a user, and it was marked by a red stripe as in \textbf{Fig.\ref{fig:layer3analysis}D}.

Then blue stripes would emerge, and the user chose one of these blue stripes as the end of the cropping frame.

Method (Ln816-822)

The density map depicts where the majority of cells were located, with yellow representing regions of high density and blue representing areas of low density. Layer \Romannum{3} of the neocortex has a lower density than layer \Romannum{2} and layer \Romannum{4}, then the ROI was specified between the two dense yellow areas for our analysis\textbf{Fig.\ref{fig:layer3analysis}D}.

After a user had clicked on the top left and the bottom right corner to define the ROI, red dashed lines appeared to show the cropping frame, see \textbf{Fig.\ref{fig:layer3analysis}D}.

This semi-automatic approach was chosen due to its reproducibility and effectiveness.

- 24. Ln 658 how is diameter in this context defined/measured?**

The diameter is now only estimated from the 2D images, as we have discovered that the inbuilt PrincipalAxisLength overestimates the diameter.

The Diameter is therefore based on the measurement from the mean segment length from the nucleator.

25. Ln 666 can the authors comment on the advantage of using this vector between centroid and furthest voxel, instead of simple the cell's major axis length (3D Feret diameter)?

Pyramidal cells have an apical dendrite pointing towards the pial surface of the brain, so our thought was that the voxel furthest away from the centroid would be located in the apical dendrite. Therefore, it made most sense in our minds to compare the furthest voxel to the centroid position to see if the radius was similar. The Feret diameter was not considered before the study, which is why we did not apply it. However, the Feret diameter seems to be a more reasonable variable to measure the longest distance within a cell. The Feret diameter will therefore be applied for the analysis and we thank the reviewer for this suggestion.

26. Ln 669 can the authors comment on choice of the "sphericity" instead of, for example, measuring "roundness" which seems more appropriate to characterize the rough edges of the cell body of pyramidal neurons?

Many compactness measures have been proposed in the literature, such as roundness, sphericity, moments, frequency descriptors, etc. Each focuses on different aspects of shapes. Using the definition of roundness as the difference between the inscribed and the circumscribed circle of a shape (<https://en.wikipedia.org/wiki/Roundness>), this measure essentially sets a lower and upper bound for the area/volume of the shape but not its circumference/surface area in 2D/3D. In contrast, many shapes with identical roundness will have quite different sphericity.

Sphericity is a more general descriptor of shape since it is also defined for objects with holes such as a torus, but the most important reasons for us to prefer sphericity are: sphericity is easy to calculate, since our software easily allows us to calculate the surface area and volume of our cells, and area and volume are fundamental descriptors of cells in their own right.

Method (Ln830-834)

Each cell's shape was assessed by approximating the sphericity, which does not have any prior assumption of shape and is independent of cell size.

The sphericity is a dimensionless ratio, and the formula is given in $\text{Eq. \ref{eq:sphericity}}$, where V and A are the volume and surface area of the segmented cell. If a cell has sphericity equal to 1, it resembles a perfect sphere \cite{Wadell1935}.

Discussion (Ln408-415)

Sphericity is valuable as a general shape descriptor since it also applies to objects having holes, such as a torus. Yet, the main reasons sphericity is used for measuring shape are as follows: sphericity is simple to compute since MATLAB easily calculates the surface area and volume of cells, and area and volume are fundamental descriptors of cells in their own right. Estimation of sphericity does not presume any shape before examining a 3D object, making it an appealing tool for differentiating between distinct forms. As a result, we implement sphericity estimates to comprehend a natural cell shape.

27. Ln675 despite having measured sphericity, the authors then proceed to classify neurons as pyramidal based on a k-means analysis, based on average radius. What is then the use of the sphericity diameter? In this line, given that we're talking about 3D shapes, perhaps circular is not the best word, "more spherical" would be better.

The reviewer is correct, we use sphericity to estimate the shape of the cells, and we assume it may be used by other researchers to equate groups of distinct neuropathologies.

The procedure has been changed in the revised article.

Method (Ln841-846)

The data used for k-means consisted of the estimated volume and sphericity in 3D. The choice of data for k-means is based on the idea that spherical objects with a smaller volume resemble smaller neurons or glial cells and hence non-pyramidal cells. For the dataset used for k-means, we decided to only use the observed cells below the average volume of the dataset to ensure that only small round cells were evaluated.\\

28. Ln689 the choice of these two values, 40um and 30um for max length considered valid for a TP detection, should be better explained, and indications given for others trying to replicate the method. Different contrast/staining results and segmentation may produce reconstructed soma that are bigger or smaller. It wasn't completely clear to me that k-means clustering itself can be used to cluster these "outlier" detections as the authors mention. This would be a more unbiased method, and one that can guarantee reproducibility (instead of a viewer manually defining a min and max size).

Previously we used k-means to remove small and big objects/cells. Now we identify big objects/cells as objects with a maximum Feret Diameter in 2D or 3D measurements which is at least three standard deviations from the mean. See Question#3, Referee#1 for a detailed response.

29. Ln708 So far I had assumed this method detected only some and at most the basal part of the dendrites (basal and apical). Here the authors assume that for cells oriented parallel to the pial surface, that the orientation parameter was not correctly measure, and I presume those cells are eliminated from the analysis. The authors mention that those were removed, however looking at the graphs for orientation, there are several cells with an orientation close to 100, in all 3 samples.

This part has been modified by using the maximum Feret diameter in order to approximate the cell orientation. The diameter of Feret gives us the longest length in a given direction, which should be towards the apical dendrite.

In the revised paper, the Feret diameter has been adopted and seems to be a much more reliable approach since no cells were above 90 degrees after implementation. We thank the reviewer for this idea and comment.

Method (Ln860-870)

Finally, the orientation vector of a pyramidal cell is defined to be the unit vector \mathbf{u} in the direction of the maximum Feret Diameter.

The maximum Feret diameter provides information about the most extended length of a cell in a particular direction, usually towards the apical dendrite. The orientation angle θ is then defined to be the angle between \mathbf{u} and the unit vector $\mathbf{u}_0 = (1, 0, 0)$ which points in the direction of the x -axis. This means that if a pyramidal cell has orientation angle 0, its orientation vector points towards the pial surface.

The orientation vector for each pyramidal cell can be calculated by

```
\begin{equation}
\mathbf{u} = \frac{\mathbf{d}-\mathbf{c}}{\|\mathbf{d}-\mathbf{c}\|},
\end{equation}
\label{eq:u}
```

where \mathbf{c} and \mathbf{d} are the vectors of the coordinates that define the maximum Feret diameter of the cell.

The orientation angle can be calculated by

```
\begin{equation}
\theta = \cos^{-1}(\mathbf{u}_0 \cdot \mathbf{u})
\end{equation}
\label{eq:ori}
```

30. Ln750 *Where is the date on tissue deformation and how was it used? I presume the authors refer to a different sample taken from a different part of the brain.*

The tissue deformation data were generated from same brain samples processed similarly before we started our study. The stereological point-counting data estimating tissue area before and after Epon processing are by the use of equation 6 translated into an estimate of tissue deformation and shown in Results. The simple methodology and data collection is explained in more detail in the referenced methods (Y Tang. et al. Age-Induced White Matter Changes in the Human Brain: A Stereological Investigation. In: Neurobiology of Aging 18.6 (Nov. 1997), pp. 609-615. doi: 10.1016/s0197-4580(97)00155-3; K-A. Dorph-Petersen et al. Tissue shrinkage and unbiased stereological estimation of particle number and size. In: Journal of Microscopy 204.3 (Dec. 2001), pp. 232-246. doi: 10.1046/j.1365-2818.2001.00958.x.)

31. Ln760 *great that the authors are sharing the data, however I would suggest uploading to Zenodo or IDR, instead of an ephemeral google drive link.*

Brilliant idea, we did not know where we could share the data, which is why we created a Google Drive account. When the paper is accepted, we will create a Zenodo account so that others can learn from the work and cite the work.

Figures and tables

32. Fig 1 in D) the legend boxes are unreadable, but instead of a yellow (as mentioned throughout the text), there is a green. In legend text where we read “were marked with a green line” perhaps replace with “green dashed line”.

The original figure is made in vector graphics and it is easy to read the text but due to the compression of the pdf file, it cannot be read here.

Legend text has now been changed to “green dashed line”.

33. Fig3 where the authors say “where each second section was in focus...” I’m not sure “focus” is the best choice of word here. I think they meant to say “section of interest”. Same happens below when the authors mention “areas in focus”. In B)iii perhaps the numbers should not have blue boxes around, it’s a bit confusing.

Thanks for the comment. It has been corrected to “section of interest”. The blue boxes have also been removed.

Figure 3

Image acquisition and 3D-stacks of aligned sections}. **(A)** Overview of a microscopic glass slide with the commercial Leica software. Systematic sampling of every third section was manually marked with three local points for the autofocus calibration. **(B)** In MATLAB, a TIF image was loaded where each second was of interest, which is different from (A) but easier to visualize as an example (i). Next, we used an entropy filter to detect sections of interest (ii). Binary masks have been computed for each section of interest and the images are ready for export (iii). The output of each exported section (iv). **(C)** Individual sections were aligned and stacked on top of each other. The stacked block of sections was then cropped down to a specific ROI (1.05x1.05 μm^2) which only contains tissue.

34. Fig4 I may have missed it, but why were those 6 lines drawn and how? What is the criterion to crop layer III to “line 6”? Also, why didn’t the authors use some unbiased quantitative parameter (instead of having a user manually user defined limit), similar to the method used in Fig1E to sample the gyrus?

One of the co-authors, Grazyna Rajkowska, is world famous for her ability to recognize and delineate BA46 in human autopsy brains, because this is very challenging. Making an automatic procedure for this is challenging, but we have now further developed our method to be semi-automatic. The lines are now removed to make the process more intuitive, and the user only needs to press on the top left corner and bottom right corner to define the ROI.

Similar question was answered in details for Question#23, Referee#2.

35. Table 5 I understand the authors decided to log the values, given that the distributions were skewed, but why was this done? It gives the impression in the graphs in Fig 5 that there is less variability than that that was actually found. Is the skewness caused by the “filtration” of unwanted cells (undesired size, shape and orientation)?

Also, the authors present a global mean that is the average of 3 averages, and present a small SD. I believe these values should be checked by a statistician, as I suspect this is not an accurate representation of the data. Perhaps this calls for nested statistics.

The skewness is not caused by the filtration but by the natural distribution of cells in general. The problem is that most do not show the histograms for the volume estimation of cells, but instead the average volume for each patient/subject, whereas a comparison is applied. However, our results have the same distribution as figure 3 in [10.1001/archpsyc.58.5.466](https://doi.org/10.1001/archpsyc.58.5.466), which looks at the BA9's deep layer III and not BA46. The deep layer III, as you can see in figure 1, is only occupied by large pyramidal cells; smaller pyramidal cells occur in the upper and middle layer and account for more cells than in the lower part. Hence, the volume of pyramidal cells varies significantly within the same layer, which produces a diverse distribution.

Also, even a slight change in the size of a neuron can have a tremendous difference in the volume estimation for larger neurons. This is because the length or radius has to be powered with 3 to estimate the volume. For example, a neuron with an average radius of 6 μm and one with a radius of 7 μm has a volume of 905 and 1436, respectively.

That is a change of 59 % in their volume, whereas there is only 16 % in their radius. That is the reason for a skewed distribution in the measured volume occurs.

Besides, here are also some histograms of the neuron size from other areas with similar distribution as ours: <https://doi.org/10.1186/s12974-014-0182-7>, [10.1016/j.brainres.2006.08.052](https://doi.org/10.1016/j.brainres.2006.08.052), [10.1523/JNEUROSCI.17-01-00372.1997](https://doi.org/10.1523/JNEUROSCI.17-01-00372.1997), <https://doi.org/10.2337/diabetes.51.3.819>.

It is statistically correct to calculate the mean for each of the three brains, compute the SD between the three brains and calculate coefficient of variation. However, because we only have three brains, it is difficult to interpret the values and no firm conclusions have been made in our paper.

36. In Fig 4, just for accuracy, there is an image showing the use of antibodies, however in the Nature's reporting summary, use of antibodies were not considered for this study.

No data has been obtained with antibodies. We just performed a practical example to show that the use of antibodies is still possible with the application of this technique on Epon sections. This is also why it was placed in supplementary materials.

37. Fig 5 “Diameter” is misspelled in the 2nd row of graphs. The use of log scales in X is unexpected. Perhaps a statistician should confirm that this is an acceptable representation of the data. It sure makes it look a lot less variable than it really is (and as can be seen in table 5 with the large SDs reported).

All graphs have been updated after code adjustments in the revised paper where both raw and log-normal transformed histograms are displayed.

Referee 3

In this manuscript, Larsen and colleagues describe a pipeline to collect and image serial semi thin sections from fixed human cortex. They next use a neural network-based method to reconstruct in 3D the cortical samples.

Main points

- 1. Overall the text is too complex, full of technical jargon and I wonder who will want to use this method, at least routinely. Importantly, at the end of the paper, what have we learnt of the human cortex cytoarchitecture that was not already known?**

This is a novel method that, for the first time, present an advanced method and pipeline for the sampling of biopsies on the complex surface of the human neocortex, embedding of biopsies in Epon, automatic sectioning and sampling of 3-400 nm thick serial sections, staining with toluidine blue, automated imaging of systematically sampled sections, alignment of sections, detection of pyramidal cells using convolutional neural network, 3D reconstruction and advanced statistical analysis of pyramidal cell properties (number, volume, diameter, sphericity, orientation) and 3D spatial distribution with cylindrical K-function. Most of these quantities of pyramidal cells in the human brain have not been published before.

There are thousands of tissue banks around the world, where tissue has been stored in fixative for years and it is difficult to use this tissue together with (neuronal) antibodies and clearing techniques and other methods (fMOST, optical microscopy, etc.).

Our method solves the problem using archive tissue and advanced quantitative analysis without the use of antibodies, even though our approach may also be combined with IHC, as shown in our paper.

This method can also be used with other archived tissue with minor modifications and is such a new pipeline for quantification of cells or other objects and their 3D interaction in general.

The point pattern analysis of neurons will help us evaluate the size/radius of columnar clustered neurons compared to a broader sample population. It is presumed that smaller sub-columns are within a macro column and will alter due to neuropathology.

- 2. First, the method only allows reconstructing a very small sample (only 700µm thick). The authors say that the biopsies contain the 6 layers of the cortex but they apparently need to remove the top and bottom layers of the cortex. Therefore, at this stage they seem unable to reconstruct a cortical column.**

As BA46 is a hotspot for research in schizophrenia and depression based on clinical studies using MRI and PET scans, we decided to focus our investigation on this region. Cellular studies have focused and found interesting differences in layer 3 of BA46, which is why other neocortical layers have been removed in this study. There is still plenty of tissue to search for columnarity/minicolumns in layer 3, and with such a focused investigation, it is easier to identify columnarity/minicolumns as mentioned in Ln96 (old version). Maybe the reviewer has

misunderstood this by writing “cortical column” or macro column, which has a diameter of ~500 µm. This is not the focus of this investigation because macro columns are already established identities in human neuroanatomy.

Introduction (Ln58-68)

The cellular organization in the human neocortex has been described as a local network of vertical columns known as 'minicolumns,' and such a column is regarded as the smallest unit capable of information processing in the cerebral cortex \cite{Mountcastle1957}. Minicolumns are radially oriented cell bodies that span through the laminar pattern perpendicular to the pial surface and can be seen using regular Nissl preparations or another cell body-revealing histological procedures.

The introduction of minicolumns was in response to studies of the patterning of apical dendrites of pyramidal cells with somata situated in layers II, III, and \cite{Fleischhauer_1972,Peters1972}. Studies have attempted to characterize and analyze the morphology of minicolumns with a 2D computerized method designed to detect subtle differences among patient groups such as schizophrenia, autism, and Alzheimer.

Introduction (Ln80-86)

BA46 was chosen since it involves working memory, attention and has been the subject of studies related to mental disorders like schizophrenia and depression \cite{Fuster1997,Selemon1998,Cruz2004,Dean2012,Udawela2017,Gibbons2009,Trojank2014,Dean2014}.

Myelinated axon bundles are potentially cortical efferents that originated in layer II/III pyramidal cells as these bundles descend to the white matter \cite{Peters1996}. In both corticocortical and thalamocortical circuitry, layer \textit{Romannum}{3} pyramidal neurons play a key role and have been suggested to be most affected by these disorders as it is the most prominent layer in BA46.

- 3. Second, the identification of the cells/neurons, the quantification of their size and polarity only relies on simple toluidine blue staining (which is not precise) and it would be more useful if they could show that is compatible with immunolabelling (the GFAP immuno, Fig S7, is quite poor).**

The GFAP in S7 is used to illustrate that immunolabelling can be performed with this procedure, however, most/all neuronal antibodies (NeuN, MAP-2, MBP, Nestin, Neurofilament _2F11, Neurofilament_DA2/FNP7/RmdO20.11) will not work in human autopsy brains placed in fixative for months/years (10.1369/jhc.7A7187.2007).

The classic Nissl technique, which stains both neurons and glial cell types, has many benefits over immunohistochemistry for quantitative experiments where cell populations must be observed. This is why we have chosen a staining method that allowed all endoplasmic reticulum (or "Nissl bodies") to be stained, followed by brain cytoarchitecture visualization, detection of pyramidal cells using convoluted neuronal networks, 3D reconstruction and advanced statistical analysis of pyramidal cell properties and 3D spatial distribution of columnarity.

The GFAP immuno image in the revised paper has been generated with a better resolution now. The manuscript was compressed from a 60MB file to less than 25MB because 25 MB is the maximum file size that could be uploaded on the website. Before publishing, the original file sizes will be uploaded to the journal's website as individual files in their original resolution.

4. Moreover, sections have many wrinkles (Fig. S3) and it is likely that their cannot be properly aligned (this should be demonstrated).

There must be an misunderstanding in relation to the alignment, since Fig.S3 was used to illustrate one scenario that can cause/generate wrinkles on sections and was not part of the analysis.

The whole point of showing S3 was to point out that not trimming all the blank resin away can cause folds due to the density difference during cutting. S3 was also not cut in optimal conditions since the indoor humidity was about 50% (figure text), compared to 80-90 % in our study, see Ln1291.

Furthermore, the picture quality of the aligned images is shown on the Github page and Fig. S12. Figures 4, 5, and 8 provide additional images of the sections used in this study.

In the revised manuscript, S3 has been changed to S9.

5. It also seems that not even all sections in the stack are imaged which again limit the interest of the pipeline.

We are using systematic uniform random sampling of sections in order to make the pipeline as efficient as possible and versatile for our specific purpose of pyramidal cell properties and 3D spatial distribution of columnarity. If the purpose changes and all sections are needed, this can easily be changed.

6. Much better methods such as expansion microscopy are now available and give access to spines and synaptic contacts.

Expansion microscopy is a very exciting technology and it can generate amazing images with fresh or lightly fixed brain tissue, however, it would be difficult/impossible to use here with human autopsy brains placed in fixative for more than 20 years. As can be seen in our paper, this methodology can also be used together with scanning electron microscopy, so it is possible to visualize spines and synaptic contacts in 3D, if the brain tissue quality is OK.

Discussion (Ln338-347)

The majority of the approaches discussed above share the use of immunolabeling to detect and localize antigens or proteins within a cell at a specific location.\\

The fundamental constraint of immunolabeling is that milder fixation conditions and a shorter detergent permeabilization incubation time are required to allow antibodies to penetrate a cell or tissue. This is particularly important for soluble proteins in the cytoplasm, which are frequently damaged or destroyed during incubation. Hence, it is challenging to create neuronal

antibodies that work in postmortem brains that have been in the fixative for months or years\cite{Lyck2007}.

Researchers have a unique opportunity as well as a terrible difficulty with archived tissue samples, enabling researchers to study disease development in great detail.

7. ***The authors mention clearing methods and claim that they do not work well with immunostaining of thick samples. However, there are many published studies, from the Ueda and Ertürk labs among others, which showed that large (centimeters) tissue pieces and even intact human organs can be immunostained and imaged in 3D.***

Thank you for the comment. We agree 100% that large tissue pieces and intact human organs can be cleared, immunostained and imaged in 3D, however, this is not the case for human autopsy brains placed in fixative for more than 20 years. This is now explained in the updated manuscript.

8. ***Figure 4: a comparison of the raw images and binary ones should be shown***

This is a very good suggestion and we have added a raw image in figure 4.

9. ***Figure 5: raw images, without pseudo colors should be shown***

We have raw images in gray scale in figure 3, figure 8 and figure S4, S12 and S16. We have raw images with toluidine blue in figure 4, figure S1, figure S9, S10, S11 and figure S18 as well as S20. We believe that is enough raw images, but will of course follow the opinion of the Editor.

10. ***Figure 6: what does it mean? Do the authors think it is self explanatory?***

Sorry for the confusion. We have updated figure 6 after improvement of the classification of the Pyramidal cells. We also agree that the analytic output from the cylindrical K-function is not self explanatory. One of our co-authors, Jesper Møller, invented the cylindrical K-function and published it in the most prestigious statistical journal Biometrika. We realize that this methodology is difficult to understand for non-statisticians, which is why the text to Figure 6 is on 171 words. Furthermore, the cylindrical K-function is explained in the “Methods” section with 527 words.

REVIEWERS' COMMENTS:

Reviewer #1 (Remarks to the Author):

The authors answered most of my concerns in the revised manuscript. Now the manuscript is commended to be published in Communications Biology.

Reviewer #2 (Remarks to the Author): 
See attachment.

**Response to Referee #2 on paper entitled:
Cellular 3D-reconstruction and analysis in Human Cerebral Cortex
using Automatic Serial Sections**

**by
Larsen NY *et al.***

We would like to sincerely thank the referees for all their suggestions for improvements and for accepting the manuscript. Like last time, we will outline changes made in response to the remaining concerns by referee #2.

Changes in the manuscript are highlighted in red.

- 1. My point was actually, what is the point of analysing "only" the shape of the soma? it is still not completely obvious to me, but I am not a neurologist.**

In relation to why only the shape of the soma is of interest, it is now clarified in the updated text in the introduction.

Introduction

A typical human neuron has thousands of complex connections with neighboring neurons, which is essential for normal function, yet the organization of these neurons is still under debate\cite{Hawkins2016}. The cellular organization in the human neocortex has been described as a local network of vertical columns containing neurons.

Neurons with similar functions are grouped together and according to different theories, these cortical columns may contain smaller columns known as minicolumns, which are the smallest unit capable of processing information in the cerebral cortex\cite{Mountcastle1957, Molnar2020}.

Cortical columns are radially oriented cell bodies that span through the laminar pattern perpendicular to the pial surface and can be seen using regular Nissl preparations or another cell body-revealing histological procedures.

The introduction of minicolumns was in response to studies of the patterning of apical dendrites of pyramidal cells with somata situated in layers \Romannum{2}, \Romannum{3}, and \Romannum{5}.

- 2. with a 0.8NA objective (assuming the condenser has a similar 0.8NA) the best resolution possible is 350-400nm. Sampling rate and actual system resolution are not the same!**
Sorry for the confusion, we have changed the text from pixel-resolution to pixel sample.

Data acquisition

Next, the microscope collected images at higher magnifications (lens 20x, NA 0.8) with pixel size down to 272 nm, which did not lose any details because the pixel size was below the expected resolution for this system's optics.

- 3. The procedure can be extremely resource consuming or even limiting, and should be evaluated by anyone who wants to try it.**

The information is already in the text: a user needs to manually export the images.

Data acquisition

The output files were named based on the positions of the glass slides in the loader, e.g., Slid1 for position 1, and the output files could be read from the commercial software Aperio ImageScope (Leica Biosystems Imaging, Inc., USA) that was part of the microscope interface. Aperio ImageScope could visualize the whole slide image and keep the high-resolution image. However, the user had to manually select a region before sections could be exported as individual image files. As a result, we built a script that could load large image files containing multiple sections and export the individual sampled sections as uncompressed TIF files in order, see

\textbf{Fig.\ref{fig:imgStacks}B}\cite{NYL}.\

4. **I'd say this in the text then...to avoid frustration from users trying to replicate this method ;)**
Using the density map is an excellent procedure for aiding any user in evaluating any ROI. This semiautomated technique also makes the entire program more adaptable if anybody wishes to measure any other areas of interest, ensuring that users are not tied into only using BA46 if it is custom-made. Layers 2 and 3 are fused in certain brain regions, whereas layer 4 is absent in others, leading directly to layer 5. Because any user may establish a ROI, the semiautomated technique allows for more flexibility in the data processing.
5. **Would it? is there evidence, or a rational for this? Given that EM is possible, i'd say the ultrastructure is not that much altered that expansion could not be attempted. Having said that, I dont think that is a reason to discard the technique described on the paper. it would be a completely different approach, and IMO one should not put the authors in the position of potentially discarding their data and method, because there is another method that "is potentially better". Discussing it, yes necessary and important for the reader. in other words, I would also encourage them to discuss even more the pros and cons of this methods vs other methods (and dont simply state that ExM or immuno are "impossible").**

Immunolabeling is one of the most difficult challenges of working with old chemical fixed tissue. Certain techniques for immunolabeling can be employed, however, they may distort or alter the tissue structure.

The Referee refers to expansion microscopy, which changes the tissue to make it larger (in all three dimensions: x, y, and z, resulting in deformation).

It is now mentioned in the discussion section that in addition to immune labeling, tissue deformation is an issue that many histological techniques may encounter. For example, avoiding tissue deformation is necessary when evaluating the size and distance between cells or organelles since you don't know, if it was generated by the histological process, illness, or a mix of both.

We don't specifically namedrop expansion microscopy in the discussion, but the text has been updated to provide more information on tissue deformation.

Discussion

The majority of the approaches discussed above share the use of immunolabeling in order to detect and localize antigens or proteins within a cell at a specific location.\\

The fundamental constraint of immunolabeling is that milder fixation conditions and a shorter detergent permeabilization incubation time are required to allow antibodies to penetrate the tissue. This is particularly important for soluble proteins in the cytoplasm, which are frequently damaged or destroyed during incubation. Hence, it is challenging to create neuronal antibodies that work in postmortem brains that have been in the fixative for months or years\cite{Lyck2007}.\\

Another well-known concern is that many histological procedures can cause tissue samples to shrink and deform, potentially altering cell

size, shape and organization. Keeping the dimensions of the tissue is especially essential if studies want to assess changes in the size and distances of any cells or organelles within the tissue. \\

Our methodology provides an effective technique for imaging smaller pieces of most archival tissue from semi-thin serial sections into a functional dataset with hardly any tissue deformation. Researchers have with our method a unique opportunity to study archived tissue samples, enabling researchers to investigate tissue and disease development in great detail. \\